# BOUNDING AND COUNTING LINEAR REGIONS OF DEEP NEURAL NETWORKS

## ABSTRACT

In this paper, we study the representational power of deep neural networks (DNN) that belong to the family of piecewise-linear (PWL) functions, based on PWL activation units such as rectifier or maxout. We investigate the complexity of such networks by studying the number of linear regions of the PWL function. Typically, a PWL function from a DNN can be seen as a large family of linear functions acting on millions of such regions. We directly build upon the work of Montúfar et al. (2014), Montúfar (2017), and Raghu et al. (2017) by refining the upper and lower bounds on the number of linear regions for rectified and maxout networks. In addition to achieving tighter bounds, we also develop a novel method to perform exact enumeration or counting of the number of linear regions with a mixed-integer linear formulation that maps the input space to output. We use this new capability to visualize how the number of linear regions change while training DNNs.

## 1 INTRODUCTION

We have witnessed an unprecedented success of deep learning algorithms in computer vision, speech, and other domains (Krizhevsky et al., 2012; Ciresan et al., 2012; Goodfellow et al., 2013; Hinton et al., 2012). While the popular deep learning architectures such as AlexNet (Krizhevsky et al., 2012), GoogleNet (Szegedy et al., 2015), and residual networks (He et al., 2016) have shown record beating performance on various image recognition tasks, empirical results still govern the design of network architecture in terms of depth and activation functions. Two important practical considerations that are part of most successful architectures are greater depth and the use of PWL activation functions such as rectified linear units (ReLUs). Due to the large gap between theory and practice, many researchers have been looking at the theoretical modeling of the representational power of DNNs (Cybenko, 1989; Anthony & Bartlett, 1999; Pascanu et al., 2014; Montúfar et al., 2014; Bianchini & Scarselli, 2014; Eldan & Shamir, 2016; Telgarsky, 2015; Mhaskar et al., 2016; Raghu et al., 2017; Montúfar, 2017).

Any continuous function can be approximated to arbitrary accuracy using a single hidden layer of sigmoid activation functions (Cybenko, 1989). This does not imply that shallow networks are sufficient to model all problems in practice. Typically, shallow networks require exponentially more number of neurons to model functions that can be modeled using much fewer activation functions in deeper ones (Delalleau & Bengio, 2011). There have been a wide variety of activation functions such as threshold ($f(z) = (z > 0)$), logistic ($f(z) = 1/(1 + \exp(-e))$), hyperbolic tangent ($f(z) = \tanh(z)$), rectified linear units (ReLUs $f(z) = \max\{0, z\}$), and maxouts ($f(z_1, z_2, \ldots, z_k) = \max\{z_1, z_2, \ldots, z_k\}$). The activation functions offer different modeling capabilities. For example, sigmoid networks are shown to be more expressive than similar-sized threshold networks (Maass et al., 1994). It was recently shown that ReLUs are more expressive than similar-sized threshold networks by deriving transformations from one network to another (Pan & Srikumar, 2016).

The complexity of neural networks belonging to the family of PWL functions can be analyzed by looking at how the network can partition the input space to an exponential number of linear response regions (Pascanu et al., 2014; Montúfar et al., 2014). The basic idea of a PWL function is simple: we can divide the input space into several regions and we have individual linear functions for each of these regions. Functions partitioning the input space to a larger number of linear regions are considered to be more complex ones, or in other words, possess better representational power. In the

case of ReLUs, it was shown that deep networks separate their input space into exponentially more linear response regions than their shallow counterparts despite using the same number of activation functions (Pascanu et al., 2014). The results were later extended and improved (Montúfar et al., 2014; Raghu et al., 2017; Montúfar, 2017; Arora et al., 2016). In particular, Montúfar et al. (2014) shows both upper and lower bounds on the maximal number of linear regions for a ReLU DNN and a single layer maxout network, and a lower bound for a maxout DNN. Furthermore, Raghu et al. (2017) and Montúfar (2017) improve the upper bound for a ReLU DNN. This upper bound asymptotically matches the lower bound from Montúfar et al. (2014) when the number of layers and input dimension are constant and all layers have the same width. Finally, Arora et al. (2016) improves the lower bound by providing a family of ReLU DNNs with an exponential number of regions given fixed size and depth.

In this work, we directly improve on the results of Montúfar et al. (Pascanu et al., 2014; Montúfar et al., 2014; Montúfar, 2017) and Raghu et al. (Raghu et al., 2017) in better understanding the representational power of DNNs employing PWL activation functions.

## 2 NOTATIONS AND BACKGROUND

We will only consider feedforward neural networks in this paper. Let us assume that the network has $n_0$ input variables given by $\mathbf{x} = \{x_1, x_2, \ldots, x_{n_0}\}$, and $m$ output variables given by $\mathbf{y} = \{y_1, y_2, \ldots, y_m\}$. Each hidden layer $l = \{1, 2, \ldots, L\}$ has $n_l$ hidden neurons whose activations are given by $\mathbf{h}^l = \{h_1^l, h_2^l, \ldots, h_{n_l}^l\}$. Let $W^l$ be the $n_l \times n_{l-1}$ matrix where each row corresponds to the weights of a neuron of layer $l$. Let $\mathbf{b}^l$ be the bias vector used to obtain the activation functions of neurons in layer $l$. Based on the $\text{ReLU}(x) = \max\{0, x\}$ activation function, the activations of the hidden neurons and the outputs are given below:

$$
\begin{aligned}
\mathbf{h}^1 &= \max\{0, W^1 \mathbf{x} + b^1\} \\
\mathbf{h}^l &= \max\{0, W^l \mathbf{h}^{l-1} + b^l\} \\
\mathbf{y} &= W^{L+1} \mathbf{h^L}
\end{aligned}
$$

As considered in Pascanu et al. (2014), the output layer is a linear layer that computes the linear combination of the activations from the previous layer without any ReLUs.

We can treat the DNN as a piecewise linear (PWL) function $F : \mathbb{R}^{n_0} \to \mathbb{R}^m$ that maps the input $\mathbf{x}$ in $\mathbb{R}^{n_0}$ to $\mathbf{y}$ in $\mathbb{R}^m$. This paper primarily deals with investigating the bounds on the linear regions of this PWL function. There are two subtly different definitions for linear regions in the literature and we will formally define them.

**Definition 1.** *Given a* PWL *function* $F : \mathbb{R}^{n_0} \to \mathbb{R}^m$, *a linear region is defined as a maximal connected subset of the input space* $\mathbb{R}^{n_0}$, *on which* $F$ *is linear (Pascanu et al., 2014; Montúfar et al., 2014).*

**Activation Pattern:** Let us consider an input vector $\mathbf{x} = \{x_1, x_2, \ldots, x_{n_0}\}$. For every layer $l$ we define an activation set $S^l \subseteq \{1, 2, \ldots, n_l\}$ such that $e \in S^l$ if and only if the ReLU $e$ is active, that is, $h_e^l > 0$. We aggregate these activation sets into a set $\mathcal{S} = (S^1, \ldots, S^l)$, which we call an activation pattern. Note that we may consider activation patterns up to a layer $l \leq L$. Activation patterns were previously defined in terms of strings (Raghu et al., 2017; Montúfar, 2017).

We say that an input $\mathbf{x}$ corresponds to an activation pattern $\mathcal{S}$ in a DNN if feeding $\mathbf{x}$ to the DNN results in the activations in $\mathcal{S}$.

**Definition 2.** *Given a* PWL *function* $F : \mathbb{R}^{n_0} \to \mathbb{R}^m$ *represented by a DNN, a linear region is the set of input vectors* $\mathbf{x}$ *that corresponds to an activation pattern* $\mathcal{S}$ *in the DNN.*

We prefer to look at linear regions as activation patterns and we interchangeably refer to $\mathcal{S}$ as an activation pattern or a region. Definitions 1 and 2 are essentially the same, except in a few degenerate cases. There could be scenarios where two different activation patterns may correspond to two adjacent regions with the same linear function. In this case, Definition 1 will produce only one linear region whereas Definition 2 will yield two linear regions. This has no effect on the bounds that we derive in this paper.

In Fig. 1(a) we show a simple ReLU DNN with two inputs $\{x_1, x_2\}$ and 3 hidden layers.

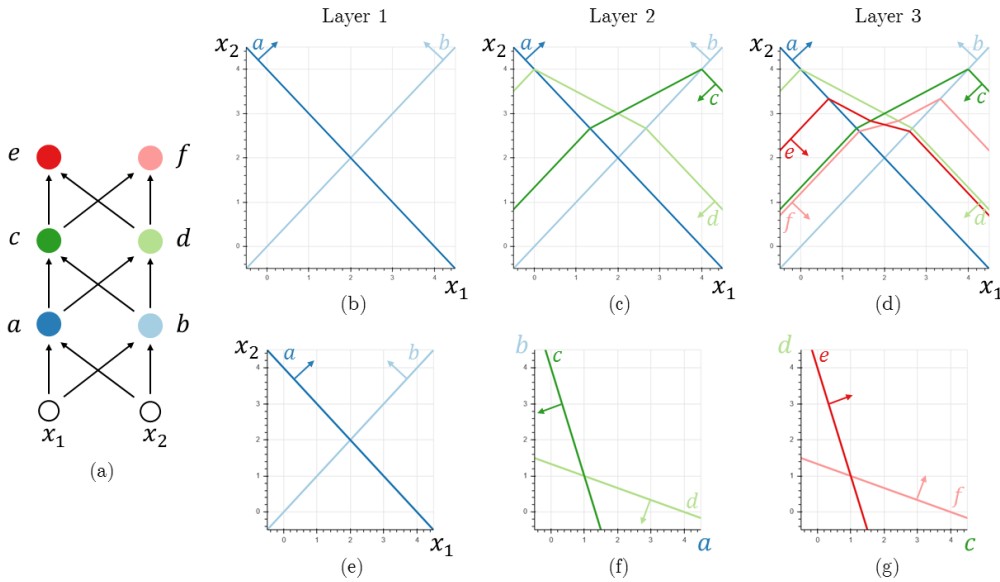

Figure 1: *(a) Simple* DNN *with two inputs and three hidden layers with 2 activation units each. (b), (c), and (d) Visualization of the hyperplanes from the first, second, and third hidden layers respectively partitioning the input space into several linear regions. The arrows indicate the directions in which the corresponding neurons are activated. (e), (f), and (g) Visualization of the hyperplanes from the first, second, and third hidden layers in the space given by the outputs of their respective previous layers.*

The activation units $\{a, b, c, d, e, f\}$ in the hidden layers can be thought of as hyperplanes that each divide the space in two. On one side of the hyperplane, the unit outputs a positive value. For all points on the other side of the hyperplane including itself, the unit outputs 0.

One may wonder: into how many regions do $n$ hyperplanes split a space? Zaslavsky (1975) shows that an arrangement of $n$ hyperplanes divides a $d$-dimensional space into at most $\sum_{s=0}^{d} \binom{n}{s}$ regions, a bound that is attained when they are in general position. The term general position basically means that a small perturbation of the hyperplanes does not change the number of regions. This corresponds to the exact maximal number of regions of a single layer DNN with $n$ ReLUs and input dimension $d$.

In Figs. 1(b)–(g), we provide a visualization of how ReLUs partition the input space. Figs. 1(e), (f), and (g) show the hyperplanes corresponding to the ReLUs at layers $l = 1, 2$, and 3 respectively. Figs. 1(b), (c), and (d) consider these same hyperplanes in the input space $x$. In Fig. 1(b), as per Zaslavsky (1975), the 2D input space is partitioned into 4 regions ($\binom{2}{0} + \binom{2}{1} + \binom{2}{2} = 4$). In Figs. 1(c) and (d), we add the hyperplanes from the second and third layers respectively, which are affected by the transformations applied in the earlier hidden layers. The regions are further partitioned as we consider additional layers.

Fig. 1 also highlights that activation boundaries behave like hyperplanes when inside a region and may bend whenever they intersect with a boundary from a previous layer. This has also been pointed out by Raghu et al. (2017). In particular, they cannot appear twice in the same region as they are defined by a single hyperplane if we fix the region. Moreover, these boundaries do not need to be connected, as illustrated in Fig. 2.

**Main Contributions**

We summarize the main contributions of this paper below:

- We achieve tighter upper and lower bounds on the maximal number of linear regions of the PWL function corresponding to a DNN that employs ReLUs. As a special case, we present the exact maximal number of regions when the input dimension is one. We ad-

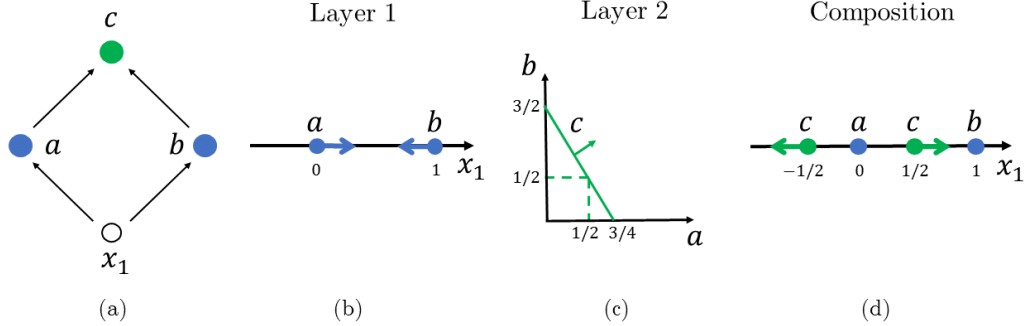

Figure 2: *(a) A network with one input $x_1$ and three activation units a, b, and c. (b) We show the hyperplanes $x_1 = 0$ and $-x_1 + 1 = 0$ corresponding to the two activation units in the first hidden layer. In other words, the activation units are given by $h_a = \max\{0, x_1\}$ and $h_b = \max\{0, -x_1 + 1\}$. (c) The activation unit in the third layer is given by $h_c = \max\{0, 4h_a + 2h_b - 3\}$. (d) The activation boundary for neuron c is disconnected.*

ditionally provide the first upper bound on the number of linear regions for multi-layer maxout networks (See Sections 3 and 4).

- We show for ReLUs that the exact maximal number of linear regions of shallow networks is larger than that of deep networks if the input dimension exceeds the number of neurons. This result is particularly interesting, since it cannot be inferred from the bounds derived in prior work.

- We use a mixed-integer linear formulation to show that exact counting of the linear regions is indeed possible. For the first time, we show the exact counting of the number of linear regions for several small-sized DNNs during the training process. This new capability can be used to evaluate the tightness of the bounds and potentially analyze the correlation between validation accuracy and the number of linear regions. It also provides new insights as to how the linear regions vary during the training process (See Section 5 and 6).

## 3 TIGHTER BOUNDS FOR RECTIFIER NETWORKS

Montúfar et al. (2014) derive an upper bound of $2^N$ for $N$ hidden units, which can be obtained by mapping linear regions to activation patterns. Raghu et al. (2017) improves this result by deriving an asymptotic upper bound of $O(n^{Ln_0})$ to the maximal number of regions, assuming $n_l = n$ for all layers $l$ and $n_0 = O(1)$. Montúfar (2017) further tightens the upper bound to $\prod_{l=1}^{L} \sum_{j=0}^{d_l} \binom{n_l}{j}$, where $d_l = \min\{n_0, n_1, \ldots, n_l\}$.

Moreover, Montúfar et al. (2014) prove a lower bound of $\left(\prod_{l=1}^{L-1} \lfloor n_l/n_0 \rfloor^{n_0}\right) \sum_{j=0}^{n_0} \binom{n_L}{j}$ when $n \geq n_0$, or asymptotically $\Omega((n/n_0)^{(L-1)n_0} n^{n_0})$. Arora et al. (2016) present a lower bound of $2 \sum_{j=0}^{n_0-1} \binom{m-1}{j} w^{L-1}$ where $2m = n_1$ and $w = n_l$ for all $l = 2, \ldots, L$. By choosing $m$ and $w$ appropriately, this lower bound is $\Omega(s^{n_0})$ where $s$ is the total size of the network. We derive both upper and lower bounds that improve upon these previous results.

### 3.1 AN UPPER BOUND ON THE NUMBER OF LINEAR REGIONS

In this section, we prove the following upper bound on the number of regions.

**Theorem 1.** *Consider a deep rectifier network with $L$ layers, $n_l$ rectified linear units at each layer $l$, and an input of dimension $n_0$. The maximal number of regions of this neural network is at most*

$$\sum_{(j_1,\ldots,j_L)\in J} \prod_{l=1}^{L} \binom{n_l}{j_l}$$

*where $J = \{(j_1,\ldots,j_L)\in\mathbb{Z}^L : 0\le j_l\le\min\{n_0,n_1-j_1,\ldots,n_{l-1}-j_{l-1},n_l\}\ \forall l=1,\ldots,L\}$. This bound is tight when $L=1$.*

Note that this is a stronger upper bound than the one that appeared in Montúfar (2017), which can be derived from this bound by relaxing the terms $n_l - j_l$ to $n_l$ and factoring the expression. When $n_0 = O(1)$ and all layers have the same width $n$, this expression has the same best known asymptotic bound $O(n^{Ln_0})$ first presented in Raghu et al. (2017).

Two insights can be extracted from the above expression:

1. **Bottleneck effect.** The bound is sensitive to the positioning of layers that are small relative to the others, a property we call the bottleneck effect. If we subtract a neuron from one of two layers with the same width, choosing the one closer to the input layer will lead to a larger (or equal) decrease in the bound. This occurs because each index $j_l$ is essentially limited by the widths of the current and previous layers, $n_0, n_1, \ldots, n_l$. In other words, smaller widths in the first few layers of the network imply a bottleneck on the bound.

   In particular for a 2-layer network, we show in Appendix A that if the input dimension is sufficiently large to not create its own bottleneck, then moving a neuron from the first layer to the second layer strictly decreases the bound, as it tightens a bottleneck.

   Figure 3a illustrates this behavior. For the solid line, we keep the total size of the network the same but shift from a small-to-large network (i.e., smaller width near the input layer and larger width near the output layer) to a large-to-small network in terms of width. We see that the bound monotonically increases as we reduce the bottleneck. If we add a layer of constant width at the end, represented by the dashed line, the bound decreases when the layers before the last become too small and create a bottleneck for the last layer.

   While this is a property of the upper bound rather than one of the exact maximal number of regions, we observe in Section 6 that empirical results for the number of regions of a trained network exhibit a behavior that resembles the bound as the width of the layers vary.

2. **Deep vs shallow for large input dimensions.** In several applications such as imaging, the input dimension can be very large. Montúfar et al. (2014) show that if the input dimension $n_0$ is constant, then the number of regions of deep networks is asymptotically larger than that of shallow (single-layer) networks. We complement this picture by establishing that if the input dimension is large, then shallow networks can attain more regions than deep networks.

   More precisely, we compare a deep network with $L$ layers of equal width $n$ and a shallow network with one layer of width $Ln$. In Appendix A, we show using Theorem 1 that if the input dimension $n_0$ exceeds the size of the network $Ln$, then the ratio between the exact maximal number of regions of the deep and of the shallow network goes to zero as $L$ approaches infinity.

   We also show in Appendix A that in a 2-layer network, if the input dimension $n_0$ is larger than both widths $n_1$ and $n_2$, then turning it into a shallow network with a layer of $n_1 + n_2$ ReLUs increases the exact maximal number of regions.

   Figure 3b illustrates this behavior. As we increase the number of layers while keeping the total size of the network constant, the bound plateaus at a value lower than the exact maximal number of regions for shallow networks. Moreover, the number of layers that yields the highest bound decreases as we increase the input dimension $n_0$.

   It is important to note that this property cannot be inferred from previous upper bounds derived in prior work, since they are at least $2^N$ when $n_0 \ge \max\{n_1,\ldots,n_L\}$, where $N$ is the total number of neurons.

   We remark that asymptotically both deep and shallow networks can attain exponentially many regions when the input dimension is at least $n$ (see Appendix B).

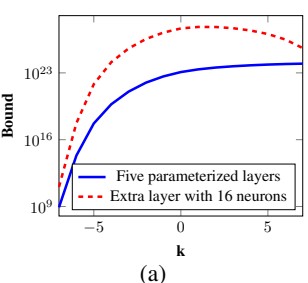 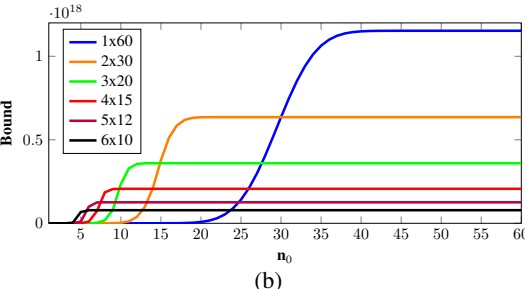

(a)                                                    (b)

Figure 3: Bounds from Theorem 1: (a) is in semilog scale, has input dimension $n_0 = 32$, and the width of the first five layers is $16 - 2k, 16 - k, 16, 16 + k, 16 + 2k$; (b) is in linear scale, evenly distributes 60 neurons in 1 to 6 layers (the single-layer case is exact), and the input dimension varies.

We now build towards the proof of Theorem 1. For a given activation set $S^l$ and a matrix $W$ with $n_l$ rows, let $\sigma_{S^l}(W)$ be the operation that zeroes out the rows of $W$ that are inactive according to $S^l$. This represents the effect of the ReLUs. For a region $S$ at layer $l - 1$, define $\bar{W}_S^l :=$ $W^l\,\sigma_{S^{l-1}}(W^{l-1})\cdots\sigma_{S^1}(W^1)$.

Each region $S$ at layer $l - 1$ may be partitioned by a set of hyperplanes defined by the neurons of layer $l$. When viewed in the input space, these hyperplanes are the rows of $\bar{W}_S^l x + b = 0$ for some $b$. To verify this, note that, if we recursively substitute out the hidden variables $h_{l-1}, \dots, h_1$ from the original hyperplane $W^l h_{l-1} + b_l = 0$ following $S$, the resulting weight matrix applied to $x$ is $\bar{W}_S^l$.

Finally, we define the dimension of a region $S$ at layer $l - 1$ as $\dim(S) := \mathrm{rank}(\sigma_{S^{l-1}}(W^{l-1})\cdots\sigma_{S^1}(W^1))$. This can be interpreted as the dimension of the space corresponding to $S$ that $W^l$ effectively partitions.

The proof of Theorem 1 focuses on the dimension of each region $S$. A key observation is that once it falls to a certain value, the regions contained in $S$ cannot recover to a higher dimension.

Zaslavsky (1975) showed that the maximal number of regions in $\mathbb{R}^d$ induced by an arrangement of $m$ hyperplanes is at most $\sum_{j=0}^d \binom{m}{j}$. Moreover, this value is attained if and only if the hyperplanes are in general position. The lemma below tightens this bound for a special case where the hyperplanes may not be in general position.

**Lemma 2.** *Consider $m$ hyperplanes in $\mathbb{R}^d$ defined by the rows of $Wx + b = 0$. Then the number of regions induced by the hyperplanes is at most $\sum_{j=0}^{\mathrm{rank}(W)} \binom{m}{j}$.*

The proof is given in Appendix C. Its key idea is that it suffices to count regions within the row space of $W$. The next lemma brings Lemma 2 into our context.

**Lemma 3.** *The number of regions induced by the $n_l$ neurons at layer $l$ within a certain region $S$ is at most $\sum_{j=0}^{\min\{n_l, \dim(S)\}} \binom{n_l}{j}$.*

*Proof.* The hyperplanes in a region $S$ of the input space are given by the rows of $\bar{W}_S^l x + b = 0$ for some $b$. By the definition of $\bar{W}_S^l$, the rank of $\bar{W}_S^l$ is upper bounded by $\min\{\mathrm{rank}(W^l), \mathrm{rank}(\sigma_{S^{l-1}}(W^{l-1})\cdots\sigma_{S^1}(W^1))\} = \min\{\mathrm{rank}(W^l), \dim(S)\}$. That is, $\mathrm{rank}(\bar{W}_S^l) \le \min\{n_l, \dim(S)\}$. Applying Lemma 2 yields the result. $\qquad\square$

In the next lemma, we show that the dimension of a region $S$ can be bounded recursively in terms of the dimension of the region containing $S$ and the number of activated neurons defining $S$.

**Lemma 4.** *Let $S$ be a region at layer $l$ and $S'$ be the region at layer $l - 1$ that contains it. Then $\dim(S) \le \min\{|S^l|, \dim(S')\}$.*

*Proof.* $\dim(S) = \mathrm{rank}(\sigma_{S^l}(W^l)\cdots\sigma_{S^1}(W^1)) \le \min\{\mathrm{rank}(\sigma_{S^l}(W^l)), \mathrm{rank}(\sigma_{S^{l-1}}(W^{l-1})\cdots$ $\sigma_{S^1}(W^1))\} \le \min\{|S^l|, \dim(S')\}$. The last inequality comes from the fact that the zeroed out rows do not count towards the rank of the matrix. $\qquad\square$

In the remainder of the proof of Theorem 1, we combine Lemmas 3 and 4 to construct a recurrence $R(l, d)$ that bounds the number of regions within a given region of dimension $d$. Simplifying this recurrence yields the expression in Theorem 1. We formalize this idea and complete the proof of Theorem 1 in Appendix D.

As a side note, Theorem 1 can be further tightened if the weight matrices are known to have small rank. The bound from Lemma 3 can be rewritten as $\sum_{j=0}^{\min\{\operatorname{rank}(W^l), \dim(\mathcal{S})\}} \binom{n_l}{j}$ if we do not relax $\operatorname{rank}(W^l)$ to $n_l$ in the proof. The term $\operatorname{rank}(W^l)$ follows through the proof of Theorem 1 and the index set $J$ in the theorem becomes $\{(j_1, \ldots, j_L) \in \mathbb{Z}^L : 0 \leq j_l \leq \min\{n_0, n_1 - j_1, \ldots, n_{l-1} - j_{l-1}, \operatorname{rank}(W^l)\} \ \forall l \geq 1\}$.

A key insight from Lemmas 3 and 4 is that the dimensions of the regions are non-increasing as we move through the layers partitioning it. In other words, if at any layer the dimension of a region becomes small, then that region will not be able to be further partitioned into a large number of regions. For instance, if the dimension of a region falls to zero, then that region will never be further partitioned. This suggests that if we want to have many regions, we need to keep dimensions high. We use this idea in the next section to construct a DNN with many regions.

## 3.2 THE CASE OF DIMENSION ONE

If the input dimension $n_0$ is equal to 1 and $n_l = n$ for all layers $l$, the upper bound presented in the previous section reduces to $(n + 1)^L$. On the other hand, the lower bound given by Montúfar et al. (2014) becomes $n^{L-1}(n+1)$. It is then natural to ask: are either of these bounds tight? The answer is that the upper bound is tight in the case of $n_0 = 1$, assuming there are sufficiently many neurons.

**Theorem 5.** *Consider a deep rectifier network with $L$ layers, $n_l \geq 3$ rectified linear units at each layer $l$, and an input of dimension 1. The maximal number of regions of this neural network is exactly $\prod_{l=1}^{L}(n_l + 1)$.*

The expression above is a simplified form of the upper bound from Theorem 1 in the case $n_0 = 1$.

The proof of this theorem in Appendix E has a construction with $n + 1$ regions that replicate themselves as we add layers, instead of $n$ as in Montúfar et al. (2014). That is motivated by an insight from the previous section: in order to obtain more regions, we want the dimension of every region to be as large as possible. When $n_0 = 1$, we want all regions to have dimension one. This intuition leads to a new construction with one additional region that can be replicated with other strategies.

## 3.3 A LOWER BOUND ON THE MAXIMAL NUMBER OF LINEAR REGIONS

Both the lower bound from Montúfar et al. (2014) and from Arora et al. (2016) can be slightly improved, since their approaches are based on extending a 1-dimensional construction similar to the one in Section 3.2. We do both since they are not directly comparable: the former bound is in terms of the number of neurons in each layer and the latter is in terms of the total size of the network.

**Theorem 6.** *The maximal number of linear regions induced by a rectifier network with $n_0$ input units and $L$ hidden layers with $n_l \geq 3n_0$ for all $l$ is lower bounded by*

$$\left(\prod_{l=1}^{L-1}\left(\left\lfloor \frac{n_l}{n_0} \right\rfloor + 1\right)^{n_0}\right) \sum_{j=0}^{n_0} \binom{n_L}{j}.$$

The proof of this theorem is in Appendix F. For comparison, the differences between the lower bound theorem (Theorem 5) from Montúfar et al. (2014) and the above theorem is the replacement of the condition $n_l \geq n_0$ by the more restrictive $n_l \geq 3n_0$, and of $\lfloor n_l/n_0 \rfloor$ by $\lfloor n_l/n_0 \rfloor + 1$.

**Theorem 7.** *For any values of $m \geq 1$ and $w \geq 2$, there exists a rectifier network with $n_0$ input units and $L$ hidden layers of size $2m + w(L - 1)$ that has $2 \sum_{j=0}^{n_0-1} \binom{m-1}{j}(w+1)^{L-1}$ linear regions.*

The proof of this theorem is in Appendix G. The differences between Theorem 2.11(i) from Arora et al. (2016) and the above theorem is the replacement of $w$ by $w + 1$. They construct a $2m$-width layer with many regions and use a one-dimensional construction for the remaining layers.

# 4 AN UPPER BOUND ON THE NUMBER OF LINEAR REGIONS FOR MAXOUT NETWORKS

We now consider a deep neural network composed of maxout units. Given weights $W_j^l$ for $j = 1, \ldots, k$, the output of a rank-$k$ maxout layer $l$ is given by

$$\mathbf{h}^l \quad = \quad \max\{W_1^l \mathbf{h}^{l-1} + b_1^l, \ldots, W_k^l \mathbf{h}^{l-1} + b_k^l\}$$

In terms of bounding number of regions, a major difference between the next result for maxout units and the previous one for ReLUs is that reductions in dimensionality due to inactive neurons with zeroed output become a particular case now. Nevertheless, using techniques similar to the ones from Section 3.1, the following theorem can be shown (see Appendix H for the proof).

**Theorem 8.** *Consider a deep neural network with $L$ layers, $n_l$ rank-$k$ maxout units at each layer $l$, and an input of dimension $n_0$. The maximal number of regions of this neural network is at most*

$$\prod_{l=1}^{L} \sum_{j=0}^{d_l} \binom{\frac{k(k-1)}{2} n_l}{j}$$

*where $d_l = \min\{n_0, n_1, \ldots, n_l\}$.*

*Asymptotically, if $n_l = n$ for all $l = 1, \ldots, L$, $n \geq n_0$, and $n_0 = O(1)$, then the maximal number of regions is at most $O((k^2 n)^{L n_0})$.*

# 5 EXACT COUNTING OF LINEAR REGIONS

If the input space $\mathbf{x} \in \mathbb{R}^{n_0}$ is bounded by minimum and maximum values along each dimension, or else if $\mathbf{x}$ corresponds to a polytope more generally, then we can define a mixed-integer linear formulation mapping polyhedral regions of $\mathbf{x}$ to the output space $\mathbf{y} \in \mathbb{R}^m$. The assumption that $\mathbf{x}$ is bounded and polyhedral is natural in most applications, where each value $x_i$ has known lower and upper bounds (e.g., the value can vary from 0 to 1 for image pixels). Among other things, we can use this formulation to count the number of linear regions.

In the formulation that follows, we use continuous variables to represent the input $\mathbf{x}$, which we can also denote as $\mathbf{h}^0$, the output of each neuron $i$ in layer $l$ as $h_i^l$, and the output $\mathbf{y}$ as $\mathbf{h}^{L+1}$. To simplify the representation, we lift this formulation to a space that also contains the output of a complementary set of neurons, each of which is active when the corresponding neuron is not. Namely, for each neuron $i$ in layer $l$ we also have a variable $\overline{h}_i^l := \max\{0, -W_i^l h^{l-1} - b_i^l\}$. We use binary variables of the form $z_i^l$ to denote if each neuron $i$ in layer $l$ is active or else if the complement of such neuron is. Finally, we assume $M$ to be a sufficiently large constant.

For a given neuron $i$ in layer $l$, the following set of constraints maps the input to the output:

$$W_i^l h^{l-1} + b_i^l = h_i^l - \overline{h}_i^l, \;\; h_i^l \leq M z_i^l, \;\; \overline{h}_i^l \leq M(1 - z_i^l), \;\; h_i^l \geq 0, \;\; \overline{h}_i^l \geq 0, \;\; z_i^l \in \{0, 1\} \qquad (1)$$

**Theorem 9.** *Provided that $|w_i^l h_j^{l-1} + b_i^l| \leq M$ for any possible value of $h^{l-1}$, a formulation with the set of constraints (1) for each neuron of a rectifier network is such that a feasible solution with a fixed value for $x$ yields the output $y$ of the neural network.*

The proof for the statement above is given in Appendix I. More details on the procedure for exact counting are in Appendix J. In addition, we show the theory for unrestricted inputs and a mixed-integer formulation for maxout networks in Appendices K and L, respectively.

These results have important consequences. First, they allow us to tap into the literature of mixed-integer representability (Jeroslow, 1987) and disjunctive programming (Balas, 1979) to understand what can be modeled on rectifier networks with a finite number of neurons and layers. To the best of our knowledge, that has not been discussed before. Second, they imply that we can use mixed-integer optimization solvers to analyze the $(\mathbf{x}, \mathbf{y})$ mapping of a trained neural network. For example, Cheng et al. (2017) use another mixed-integer formulation to generate adversarial examples of a DNN. That is technically feasible due to the linear proportion between the size of the neural network and that of the mixed-integer formulation. Compared to Cheng et al. (2017), we show in Appendix I that formulation (1) can be implemented with further refinements on the value of the $M$ constants.

## 6 EXPERIMENTS

We perform two different experiments for region counting using small-sized networks with ReLU activation units on the MNIST benchmark dataset (LeCun et al., 1998). In the first experiment, we generate rectifier networks with 1 to 4 hidden layers having 10 neurons each, with final test error between 6 and 8%. The training was carried out for 20 epochs or training steps, and we count the number of linear regions during each training step. For those networks, we count the number of linear regions within $0 \leq x \leq 1$ in which a single neuron is active in the output layer, hence partitioning these regions in terms of the digits that they classify. In Fig. 4, we show how the number of regions classifying each digit progresses during training. Some digits have zero linear regions in the beginning, which explains why they begin later in the plot. The total number of such regions per training step is presented in Fig. 5(a) and error measures are found in Appendix M. Overall, we observe that the number of linear regions jumps orders of magnitude are varies more widely for each added layer. Furthermore, there is an initial jump in the number of linear regions classifying each digit that seems proportional to the number of layers.

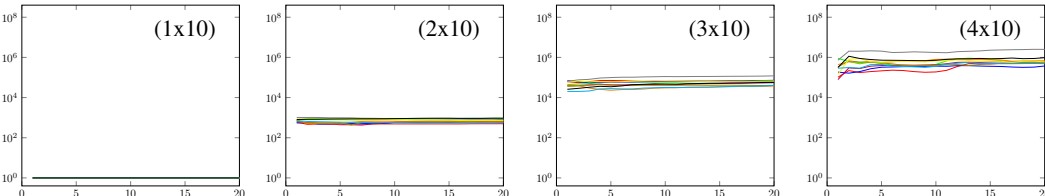

Figure 4: *Total number of regions classifying each digit (different colors for 0-9) of MNIST alone as training progresses, each plot corresponding to a different number of hidden layers.*

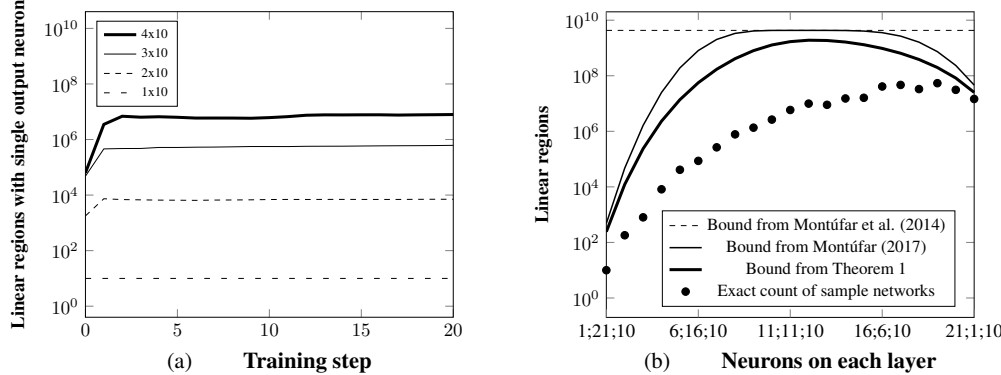

Figure 5: *(a) Total number of linear regions classifying a single digit of MNIST as training progresses, each plot corresponding to a different number of hidden layers. (b) Comparison of upper bounds from Montúfar et al. (2014), Montúfar (2017), and from Theorem 1 with the total number of linear regions of a network with two hidden layers totaling 22 neurons.*

In the second experiment, we train rectifier networks with two hidden layers summing up to 22 neurons. We train a network for each width configuration under the same conditions as above, with the test error in half of them ranging from 5 to 6%. In this case, we count all linear regions within $0 \leq x \leq 1$, hence not restricting by activation in output layer as before. The number of linear regions of these networks are plotted in Fig. 5(b), along with the upper bound from Theorem 1 and the upper bounds from Montúfar et al. (2014) and Montúfar (2017). Error measures of both experiments can be found in Appendix M and runtimes for counting the linear regions in Appendix N.

## 7 DISCUSSION

The representational power of a DNN can be studied by observing the number of linear regions of the PWL function that the DNN represents. In this work, we improve on the upper and lower bounds on the linear regions for rectified networks derived in prior work (Montúfar et al., 2014; Raghu et al., 2017; Montúfar, 2017; Arora et al., 2016) and introduce a first upper bound for multi-layer maxout networks. We obtain several valuable insights from our extensions.

Our ReLU upper bound indicates that small widths in early layers cause a bottleneck effect on the number of regions. If we reduce the width of an early layer, the dimensions of the linear regions become irrecoverably smaller throughout the network and the regions will not be able to be partitioned as much. Moreover, the dimensions of the linear regions are not only driven by width, but also the number of activated ReLUs corresponding to the region. This intuition allowed us to create a 1-dimensional construction with the maximal number of regions by eliminating a zero-dimensional bottleneck. An unexpected and useful consequence of our result is that shallow networks can attain more linear regions when the input dimensions exceed the number of neurons of the DNN.

In addition to achieving tighter bounds, we use a mixed-integer linear formulation that maps the input space to the output to show the exact counting of the number of linear regions for several small-sized DNNs during the training process. In the first experiment, we observed that the number of linear regions correctly classifying each digit of the MNIST benchmark increases and vary in proportion to the depth of the network during the first training epochs. In the second experiment, we count the total number of linear regions as we vary the width of two layers with a fixed number of neurons, and we experimentally validate the bottleneck effect by observing that the results follow a similar pattern to the upper bound that we show.

Our current results suggest new avenues for future research. First, we believe that the study of linear regions may eventually lead to insights in how to design better DNNs in practice, for example by further validating the bottleneck effect found in this study. Other properties of the bounds may turn into actionable insights if confirmed as these bounds get sufficiently close to the actual number of regions. For example, the plots in Appendix O show that there are particular network depths that maximize our ReLU upper bound for a given input dimension and number of neurons. In a sense, the number of neurons is a proxy to the computational resources available. We also believe that analyzing the shape of the linear regions is a promising idea for future work, which could provide further insight in how to design DNNs. Another important line of research is to understand the exact relation between the number of linear regions and accuracy, which may also involve the potential for overfitting. We conjecture that the network training is not likely to generalize well if there are so many regions that each point can be singled out in a different region, in particular if regions with similar labels are unlikely to be compositionally related. Second, applying exact counting to larger networks would depend on more efficient algorithms or on using approximations instead. In any case, the exact counting at a smaller scale can assess the quality of the current bounds and possibly derive insights for tighter bounds in future work, hence leading to insights that could be scaled up.

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

# Appendices

Most of the proofs for theorems and lemmas associated with the upper and lower bounds on the linear regions are provided below. The theory for mixed-integer formulation for exact counting in the case of maxouts and unrestricted inputs are also provided below.

## A  ANALYSIS OF THE BOUND FROM THEOREM 1

In this section, we present properties of the upper bound for the number of regions of a rectifier network from Theorem 1. Denote the bound by $B(n_0, n_1, \ldots, n_L)$, where $n_0$ is the input dimension and $n_1, \ldots, n_L$ are the widths of layers 1 through $L$ of the network. That is,

$$B(n_0, n_1, \ldots, n_L) := \sum_{(j_1, \ldots, j_L) \in J} \prod_{l=1}^{L} \binom{n_l}{j_l}$$

Instead of expressing $J$ as in Theorem 1, we rearrange it to a more convenient form for the proofs in this section:

$$J = \{(j_1, \ldots, j_L) \in \mathbb{Z}^L : j_l + j_k \leq n_k \ \forall k = 1, \ldots, l-1 \ \forall l = 2, \ldots, L$$
$$j_l \leq n_0 \ \forall l = 1, \ldots, L$$
$$0 \leq j_l \leq n_l \ \forall l = 1, \ldots, L\}.$$

Note that whenever we assume $n_0 \geq \max\{n_1, \ldots, n_l\}$, then the bound inequality for $n_0$ becomes redundant and can be removed.

Some of the results have implications in terms of the exact maximal number of regions. We denote it by $R(n_0, n_1, \ldots, n_L)$, following the same notation above.

Moreover, the following lemma is useful throughout the section.

**Lemma 10.**

$$\sum_{j=0}^{k} \binom{n_1 + \ldots + n_L}{j} = \sum_{\substack{j_1 + \ldots + j_L \leq k \\ 0 \leq j_l \leq n_l \ \forall l}} \binom{n_1}{j_1} \binom{n_2}{j_2} \cdots \binom{n_L}{j_L}.$$

*Proof.* The result comes from taking a generalization of Vandermonde's identity and adding the summation of $j$ from 0 to $k$ as above. $\square$

We first examine some properties related to 2-layer networks. The proposition below characterizes the bound when $L = 2$ for large input dimensions.

**Proposition 11.** *Consider a 2-layer network with widths $n_1, n_2$ and input dimension $n_0 \geq n_1$ and $n_0 \geq n_2$. Then*

$$B(n_0, n_1, n_2) = \sum_{j=0}^{n_1} \binom{n_1 + n_2}{j}$$

*If $n_0 < n_1$ or $n_0 < n_2$, the above holds with inequality: $B(n_0, n_1, n_2) \leq \sum_{j=0}^{n_1} \binom{n_1 + n_2}{j}$.*

*Proof.* If $n_0 \geq n_1$ and $n_0 \geq n_2$, the bound inequalities for $n_0$ in the index set $J$ become redundant. By applying Lemma 10, we obtain

$$B(n_0, n_1, n_2) = \sum_{\substack{0 \leq j_1 + j_2 \leq n_1 \\ 0 \leq j_l \leq n_l \ \forall l}} \binom{n_1}{j_1} \binom{n_2}{j_2} = \sum_{j=0}^{n_1} \binom{n_1 + n_2}{j}.$$

If $n_0 < n_1$ or $n_0 < n_2$, then its index set $J$ is contained by the one above, and thus the first equal sign above becomes a less-or-equal sign. $\square$

Recall that the expression on the right-hand side of Proposition 11 is equal to the maximal number of regions of a single-layer network with $n_1 + n_2$ ReLUs and input dimension $n_1$, as discussed in Section 2. Hence, the proposition implies that for large input dimensions, a two-layer network has no more regions than a single-layer network with the same number of neurons, as formalized below.

**Corollary 12.** *Consider a 2-layer network with widths $n_1, n_2 \geq 1$ and input dimension $n_0 \geq n_1$ and $n_0 \geq n_2$. Then $R(n_0, n_1, n_2) \leq R(n_0, n_1 + n_2)$.*

*Moreover, this inequality is strict when $n_0 > n_1$.*

*Proof.* This is a direct consequence of Proposition 11:

$$R(n_0, n_1, n_2) \leq B(n_0, n_1, n_2) = \sum_{j=0}^{n_1} \binom{n_1 + n_2}{j} \leq \sum_{j=0}^{n_0} \binom{n_1 + n_2}{j} = R(n_0, n_1 + n_2).$$

Note that if $n_0 > n_1$, then the second inequality can be turned into a strict inequality.  □

The next corollary illustrates the bottleneck effect for two layers. It states that for large input dimensions, moving a neuron from the second layer to the first strictly increases the bound.

**Corollary 13.** *Consider a 2-layer network with widths $n_1, n_2$ and input dimension $n_0 \geq n_1 + 1$ and $n_0 \geq n_2 + 1$. Then $B(n_0, n_1 + 1, n_2) > B(n_0, n_1, n_2 + 1)$.*

*Proof.* By Proposition 11,

$$B(n_0, n_1 + 1, n_2) = \sum_{j=0}^{n_1+1} \binom{(n_1 + 1) + n_2}{j} > \sum_{j=0}^{n_1} \binom{n_1 + (n_2 + 1)}{j} = B(n_0, n_1, n_2 + 1).$$

□

The assumption that $n_0$ must be large is required for the above proposition; otherwise, the input itself may create a bottleneck with respect to the second layer as we decrease its size. Note that the bottleneck affects all subsequent layers, not only the layer immediately after it.

However, it is not true that moving neurons to earlier layers always increases the bound. For instance, with three layers, $B(4, 3, 2, 1) = 47 > 46 = B(4, 4, 1, 1)$.

In the remainder of this section, we consider deep networks of equal widths $n$. The next proposition can be viewed as an extension of Proposition 11 for multiple layers. It states that for a network with widths and input dimension $n$ and at least 4 layers, if we halve the number of layers and redistribute the neurons so that the widths become $2n$, then the bound increases. In other words, if we assume the bound to be close to the maximal number of regions, it suggests that making a deep network shallower allows for more regions when the input dimension is equal to the width.

**Proposition 14.** *Consider a $2L$-layer network with equal widths $n$ and input dimension $n_0 = n$. Then*

$$B(n, \underbrace{n, \ldots, n}_{2L \text{ times}}) \leq B(n, \underbrace{2n, \ldots, 2n}_{L \text{ times}}).$$

*This inequality is met with equality when $L = 1$ and strict inequality when $L \geq 2$.*

*Proof.* When $n_0 = n$, the inequalities $j_l \leq \min\{n_0, 2n - j_1, \ldots, 2n - j_{l-1}, 2n\}$ appearing in $J$ (in the form presented in Theorem 1) can be simplified to $j_l \leq n$. Therefore, using Lemma 10, the bound on the right-hand side becomes

$$B(n, \underbrace{2n, \ldots, 2n}_{L \text{ times}}) = \sum_{j_1=0}^{n} \sum_{j_2=0}^{n} \cdots \sum_{j_L=0}^{n} \prod_{l=1}^{L} \binom{2n}{j_l} = \left( \sum_{j=0}^{n} \binom{2n}{j} \right)^L = \left( \sum_{j_1=0}^{n} \sum_{j_2=0}^{n-j_1} \binom{n}{j_1}\binom{n}{j_2} \right)^L$$

$$\geq \sum_{(j_1, \ldots, j_{2L}) \in J} \prod_{l=1}^{2L} \binom{n}{j_l} = B(n, \underbrace{n, \ldots, n}_{2L \text{ times}}).$$

where $J$ above is the index set from Theorem 1 applied to $n_0 = n_l = n$ for all $l = 1, \ldots, 2L$. Note that we can turn the inequality into equality when $L = 1$ (also becoming a consequence of Proposition 11) and into strict inequality when $L \geq 2$. $\qquad \square$

Next, we provide an upper bound that is independent of $n_0$.

**Proposition 15.** *Consider an $L$-layer network with equal widths $n$ and any input dimension $n_0 \geq 0$.*

$$B(n_0, n, \ldots, n) \leq 2^{Ln} \left( \frac{1}{2} + \frac{1}{2\sqrt{\pi n}} \right)^{L/2} \sqrt{2}$$

*Proof.* Since we are deriving an upper bound, we can assume $n_0 \geq n$, as the bound is nondecreasing on $n_0$. We first assume that $L$ is even. We relax some of the constraints of the index set $J$ from Theorem 1 and apply Vandermonde's identity on each pair:

$$B(n_0, n, \ldots, n) \leq \sum_{j_1=0}^{n} \sum_{j_2=0}^{n-j_1} \binom{n}{j_1}\binom{n}{j_2} \sum_{j_3=0}^{n} \sum_{j_4=0}^{n-j_3} \binom{n}{j_3}\binom{n}{j_4} \cdots \sum_{j_{L-1}=0}^{n} \sum_{j_L=0}^{n-j_{L-1}} \binom{n}{j_{L-1}}\binom{n}{j_L}$$

$$= \left( \sum_{j=0}^{n} \binom{2n}{j} \right)^{L/2} = \left( \frac{2^{2n} + \binom{2n}{n}}{2} \right)^{L/2} \leq \left( \frac{2^{2n} + \frac{2^{2n}}{\sqrt{\pi n}}}{2} \right)^{L/2}$$

$$= 2^{Ln} \left( \frac{1}{2} + \frac{1}{2\sqrt{\pi n}} \right)^{L/2}.$$

The bound on $\binom{2n}{n}$ is a direct application of Stirling's approximation (Stirling, 1730). If $L$ is odd, then we can write

$$B(n_0, n, \ldots, n) \leq \left( \sum_{j=0}^{n} \binom{2n}{j} \right)^{(L-1)/2} \left( \sum_{j=0}^{n} \binom{n}{j} \right) = \left( \sum_{j=0}^{n} \binom{2n}{j} \right)^{L/2} \frac{2^n}{\left( \sum_{j=0}^{n} \binom{2n}{j} \right)^{1/2}}$$

$$\leq \left( \sum_{j=0}^{n} \binom{2n}{j} \right)^{L/2} \frac{2^n}{(2^{2n}/2)^{1/2}} = \left( \sum_{j=0}^{n} \binom{2n}{j} \right)^{L/2} \sqrt{2}$$

$$\leq 2^{Ln} \left( \frac{1}{2} + \frac{1}{2\sqrt{\pi n}} \right)^{L/2} \sqrt{2}$$

where the last inequality is analogous to the even case. Hence, the result follows. $\qquad \square$

**Corollary 16.** *Consider an $L$-layer network with equal widths $n$ and any input dimension $n_0 \geq 0$.*

$$\lim_{L \to \infty} \frac{R(n_0, n, \ldots, n)}{2^{Ln}} = 0$$

*Proof.* By Proposition 15 and Theorem 1, the ratio between $R(n_0, n, \ldots, n)$ and $2^{Ln}$ is at most $\sqrt{\frac{1}{2} + \frac{1}{2\sqrt{\pi n}}}^{L} \sqrt{2}$. Since the base of the first term is less than 1 for all $n \geq 1$ and $\sqrt{2}$ is a constant, the ratio goes to 0 as $L$ goes to infinity. $\qquad \square$

In particular, Corollary 16 implies that if $n_0$ exceeds the total size of the network, that is, $n_0 \geq Ln$, then $\lim_{L \to \infty} \frac{R(n_0, n, \ldots, n)}{R(n_0, Ln)} = 0$. In other words, the ratio between the maximal number of regions of a deep network and a shallow network goes to zero as $L$ goes to infinity.

## B  EXPONENTIAL MAXIMAL NUMBER OF REGIONS WHEN INPUT DIMENSION IS LARGE

**Proposition 17.** *Consider an $L$-layer rectifier network with equal widths $n$ and input dimension $n_0 \geq n/3$. Then the maximal number of regions is $\Omega(2^{\frac{2}{3}Ln})$.*

*Proof.* It suffices to show that a lower bound such as the one from Theorem 6 grows exponentially large. For simplicity, we consider the lower bound $(\prod_{l=1}^{L}(\lfloor n_l/n_0 \rfloor + 1))^{n_0}$, which is the bound obtained before the last tightening step in the proof of Theorem 6 (see Appendix F).

Note that replacing $n_0$ in the above expression by a value $n_0'$ smaller than the input dimension still yields a valid lower bound. This holds because increasing the input dimension of a network from $n_0'$ to $n_0$ cannot decrease its maximal number of regions.

Choose $n_0' = \lfloor n/3 \rfloor$, which satisfies $n_0' \leq n_0$ and the condition $n \geq 3n_0'$ of Theorem 6. The lower bound can be expressed as $(\lfloor n/\lfloor n/3 \rfloor \rfloor + 1)^{L\lfloor n/3 \rfloor} \geq 4^{L\lfloor n/3 \rfloor}$. This implies that the maximal number of regions is $\Omega(2^{\frac{2}{3}Ln})$. □

## C  PROOF OF LEMMA 2

**Lemma 2.** *Consider $m$ hyperplanes in $\mathbb{R}^d$ defined by the rows of $Wx + b = 0$. Then the number of regions induced by the hyperplanes is at most $\sum_{j=0}^{\text{rank}(W)} \binom{m}{j}$.*

*Proof.* Consider the row space $\mathcal{R}(W)$ of $W$, which is a subspace of $\mathbb{R}^d$ of dimension $\text{rank}(W)$. We show that the number of regions $N_{\mathbb{R}^d}$ in $\mathbb{R}^d$ is equal to the number of regions $N_{\mathcal{R}(W)}$ in $\mathcal{R}(W)$ induced by $Wx + b = 0$ restricted to $\mathcal{R}(W)$. This suffices to prove the lemma since $\mathcal{R}(W)$ has at most $\sum_{j=0}^{\text{rank}(W)} \binom{m}{j}$ regions according to Zaslavsky's theorem.

Since $\mathcal{R}(W)$ is a subspace of $\mathbb{R}^d$, it directly follows that $N_{\mathcal{R}(W)} \leq N_{\mathbb{R}^d}$. To show the converse, we apply the orthogonal decomposition theorem from linear algebra: any point $\bar{x} \in \mathbb{R}^d$ can be expressed uniquely as $\bar{x} = \hat{x} + y$, where $\hat{x} \in \mathcal{R}(W)$ and $y \in \mathcal{R}(W)^\perp$. Here, $\mathcal{R}(W)^\perp = \text{Ker}(W) := \{y \in \mathbb{R}^d : Wy = 0\}$, and thus $W\bar{x} = W\hat{x} + Wy = W\hat{x}$. This means $\bar{x}$ and $\hat{x}$ lie on the same side of each hyperplane of $Wx = b$ and thus belong to the same region. In other words, given any $\bar{x} \in \mathbb{R}^d$, its region is the same one that $\hat{x} \in \mathcal{R}(W)$ lies in. Therefore, $N_{\mathbb{R}^d} \leq N_{\mathcal{R}(W)}$. Hence, $N_{\mathbb{R}^d} = N_{\mathcal{R}(W)}$ and the result follows. □

## D  PROOF OF THEOREM 1

**Theorem 1.** *Consider a deep rectifier network with $L$ layers, $n_l$ rectified linear units at each layer $l$, and an input of dimension $n_0$. The maximal number of regions of this neural network is at most*

$$\sum_{(j_1,\ldots,j_L) \in J} \prod_{l=1}^{L} \binom{n_l}{j_l}$$

*where $J = \{(j_1,\ldots,j_L) \in \mathbb{Z}^L : 0 \leq j_l \leq \min\{n_0, n_1 - j_1,\ldots,n_{l-1} - j_{l-1}, n_l\} \; \forall l = 1,\ldots,L\}$. This bound is tight when $L = 1$.*

*Proof.* As illustrated in Figure 1, the partitioning can be viewed as a sequential process: at each layer, we partition the regions obtained from the previous layer. When viewed in the input space, each region $\mathcal{S}$ obtained at layer $l - 1$ is potentially partitioned by $n_l$ hyperplanes given by the rows of $\bar{W}_{\mathcal{S}}^l + b = 0$ for some bias $b$. Some of these hyperplanes may fall outside the interior of $\mathcal{S}$ and do not partition the region.

With this process in mind, we recursively bound the number of subregions within a region. More precisely, we construct a recurrence $R(l, d)$ to be an upper bound to the maximal number of regions obtained from partitioning a region of dimension $d$ with layers $l, l + 1, \ldots, L$. The base case of

the recurrence is given by Lemma 3: $R(L, d) = \sum_{j=0}^{\min\{n_L, d\}} \binom{n_L}{j}$. Based on Lemma 4, we can write the recurrence by grouping together regions with the same activation set size $|S^l|$, as follows: $R(l, d) = \sum_{j=0}^{n_l} N_{n_l, d, j} R(l + 1, \min\{j, d\})$ for all $l = 1, \ldots, L - 1$. Here, $N_{n_l, d, j}$ represents the maximum number of regions with $|S^l| = j$ obtained by partitioning a space of dimension $d$ with $n_l$ hyperplanes. We bound this value next.

For each $j$, there are at most $\binom{n_l}{j}$ regions with $|S^l| = j$, as they can be viewed as subsets of $n_l$ neurons of size $j$. In total, Lemma 3 states that there are at most $\sum_{j=0}^{\min\{n_l, d\}} \binom{n_l}{j}$ regions. If we allow these regions to have the highest $|S^l|$ possible, for each $j$ from 0 to $\min\{n_l, d\}$ we have at most $\binom{n_l}{n_l - j} = \binom{n_l}{j}$ regions with $|S^l| = n_l - j$.

Therefore, we can write the recurrence as

$$R(l, d) = \begin{cases} \displaystyle\sum_{j=0}^{\min\{n_l, d\}} \binom{n_l}{j} R(l + 1, \min\{n_l - j, d\}) & \text{if } 1 \leq l \leq L - 1, \\[2em] \displaystyle\sum_{j=0}^{\min\{n_L, d\}} \binom{n_L}{j} & \text{if } l = L. \end{cases}$$

The recurrence $R(1, n_0)$ can be unpacked to

$$\sum_{j_1=0}^{\min\{n_1, d_1\}} \binom{n_1}{j_1} \sum_{j_2=0}^{\min\{n_2, d_2\}} \binom{n_2}{j_2} \ldots \sum_{j_L=0}^{\min\{n_L, d_L\}} \binom{n_L}{j_L}$$

where $d_l = \min\{n_0, n_1 - j_1, \ldots, n_{l-1} - j_{l-1}\}$. This can be made more compact, resulting in the final expression.

The bound is tight when $L = 1$ since it becomes $\sum_{j=0}^{\min\{n_0, n_1\}} \binom{n_1}{j}$, which is the maximal number of regions of a single-layer network. $\qquad\square$

## E    PROOF OF THEOREM 5

**Theorem 5.** *Consider a deep rectifier network with $L$ layers, $n_l \geq 3$ rectified linear units at each layer $l$, and an input of dimension 1. The maximal number of regions of this neural network is exactly $\prod_{l=1}^{L} (n_l + 1)$.*

*Proof.* Section 3 provides us with a helpful insight to construct an example with a large number of regions. It tells us that we want regions to have large dimension in general. In particular, regions of dimension zero cannot be further partitioned. This suggests that the one-dimensional construction from Montúfar et al. (2014) can be improved, as it contains $n$ regions of dimension one and 1 region of dimension zero. This is because all ReLUs point to the same direction as depicted in Fig. 6, leaving one region with an empty activation pattern.

Our construction essentially increases the dimension of this region from zero to one. This is done by shifting the neurons forward and flipping the direction of the third neuron, as illustrated in Fig. 6. We assume $n \geq 3$.

We review the intuition behind the construction strategy from Montúfar et al. (2014). They construct a linear function $\tilde{h} : \mathbb{R} \to \mathbb{R}$ with a zigzag pattern from $[0, 1]$ to $[0, 1]$ that is composed of $n$ ReLUs. More precisely, $\tilde{h}(x) = (1, -1, 1, \ldots, \pm 1)^\top (h_1(x), h_2(x), \ldots, h_n(x))$, where $h_i(x)$ for $i = 1, \ldots, n$ are ReLUs. This linear function can be absorbed in the preactivation function of the next layer.

The zigzag pattern allows it to replicate in each slope a scaled copy of the function in the domain $[0, 1]$. Fig. 7 shows an example of this effect. Essentially, when we compose $\tilde{h}$ with itself, each linear piece in $[t_1, t_2]$ such that $\tilde{h}(t_1) = 0$ and $\tilde{h}(t_2) = 1$ maps the entire function $\tilde{h}$ to the interval $[t_1, t_2]$, and each piece such that $\tilde{h}(t_1) = 1$ and $\tilde{h}(t_2) = 2$ does the same in a backward manner.

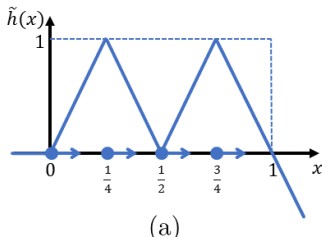 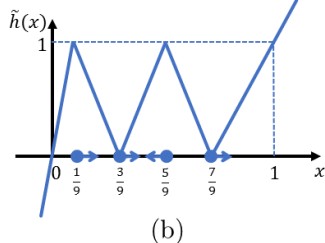

Figure 6: (a) The 1D construction from Montúfar et al. (2014). All units point to the right, leaving a region with dimension zero before the origin. (b) The 1D construction described in this section. Within the interval $[0, 1]$ there are five regions instead of the four in (a).

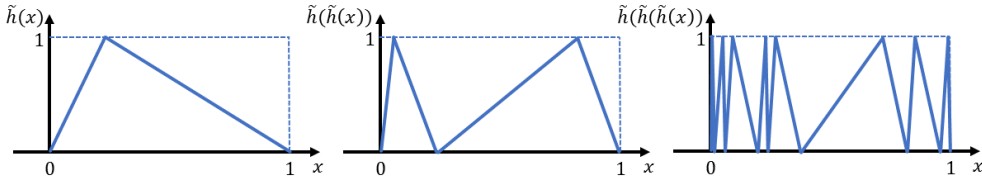

Figure 7: A function with a zigzag pattern composed with itself. Note that the entire function is replicated within each linear region, up to a scaling factor.

In our construction, we want to use $n$ ReLUs to create $n + 1$ regions instead of $n$. In other words, we want the construct this zigzag pattern with $n + 1$ slopes. In order to do that, we take two steps to give ourselves more freedom. First, observe that we only need each linear piece to go from zero to one or one to zero; that is, the construction works independently of the length of each piece. Therefore, we turn the breakpoints into parameters $t_1, t_2, \ldots, t_n$, where $0 < t_1 < t_2 < \ldots < t_n < 1$. Second, we add sign and bias parameters to the function $\tilde{h}$. That is, $\tilde{h}(x) = (s_1, s_2, \ldots, s_n)^\top (h_1(x), h_2(x), \ldots, h_n(x)) + d$, where $s_i \in \{-1, +1\}$ and $d$ are parameters to be set. Here, $h_i(x) = \max\{0, \tilde{w}_i x + \tilde{b}_i\}$ since it is a ReLU.

We define $w_i = s_i \tilde{w}_i$ and $b_i = s_i \tilde{b}_i$, which are the weights and biases we seek in each interval to form the zigzag pattern. The parameters $s_i$ are needed because the signs of $\tilde{w}_i$ cannot be arbitrary: it must match the directions the ReLUs point towards. In particular, we need a positive slope ($\tilde{w}_i > 0$) if we want $i$ to point right, and a negative slope ($\tilde{w}_i < 0$) if we want $i$ to point left. Hence, without loss of generality, we do not need to consider the $s_i$'s any further since they will be directly defined from the signs of the $w_i$'s and the directions. More precisely, $s_i = 1$ if $w_i \geq 0$ and $s_i = -1$ otherwise for $i = 1, 2, 4, \ldots, n$, and $s_3 = -1$ if $w_3 \geq 0$ and $s_3 = 1$ otherwise.

To summarize, our parameters are the weights $w_i$ and biases $b_i$ for each ReLU, a global bias $d$, and the breakpoints $0 < t_1 < \ldots < t_n < 1$. Our goal is to find values for these parameters such that each piece in the function $\tilde{h}$ with domain in $[0, 1]$ is linear from zero to one or one to zero.

More precisely, if the domain is $[s, t]$, we want each linear piece to be either $\frac{1}{t-s}x - \frac{s}{t-s}$ or $-\frac{1}{t-s}x + \frac{t}{t-s}$, which define linear functions from zero to one and from one to zero respectively. Since we want a zigzag pattern, the former should happen for the interval $[t_i, t_{i-1}]$ when $i$ is odd and the latter should happen when $i$ is even.

There is one more set of parameters that we will fix. Each ReLU corresponds to a hyperplane, or a point in dimension one. In fact, these points are the breakpoints $t_1, \ldots, t_n$. They have directions that define for which inputs the neuron is activated. For instance, if a neuron $h_i$ points to the right, then the neuron $h_i(x)$ outputs zero if $x \leq t_i$ and the linear function $w_i x + b_i$ if $x > t_i$.

As previously discussed, in our construction all neurons point right except for the third neuron $h_3$, which points left. This is to ensure that the region before $t_1$ has one activated neuron instead of zero, which would happen if all neurons pointed left. However, although ensuring every region has dimension one is necessary to reach the bound, not every set of directions yields valid weights. These directions are chosen so that they admit valid weights.

The directions of the neurons tells us which neurons are activated in each region. From left to right, we start with $h_3$ activated, then we activate $h_1$ and $h_2$ as we move forward, we deactivate $h_3$, and finally we activate $h_4, \ldots, h_n$ in sequence. This yields the following system of equations, where $t_{n+1}$ is defined as 1 for simplicity:

$$w_3 x + (b_3 + d) = \frac{1}{t_1} x \qquad (R_1)$$

$$(w_1 + w_3) x + (b_1 + b_3 + d) = -\frac{1}{t_2 - t_1} x + \frac{t_2}{t_2 - t_1} \qquad (R_2)$$

$$(w_1 + w_2 + w_3) x + (b_1 + b_2 + b_3 + d) = \frac{1}{t_3 - t_2} x - \frac{t_2}{t_3 - t_2} \qquad (R_3)$$

$$(w_1 + w_2) x + (b_1 + b_2 + d) = -\frac{1}{t_4 - t_3} x + \frac{t_4}{t_4 - t_3} \qquad (R_4)$$

$$\left( w_1 + w_2 + \sum_{j=4}^{i-1} w_j \right) x + \left( b_1 + b_2 + \sum_{j=4}^{i-1} b_j + d \right) = \begin{cases} \frac{1}{t_i - t_{i-1}} x - \frac{t_{i-1}}{t_i - t_{i-1}} & \text{if } i \text{ is odd} \\ -\frac{1}{t_i - t_{i-1}} x + \frac{t_i}{t_i - t_{i-1}} & \text{if } i \text{ is even} \end{cases}$$

$$(R_i)$$

$$\text{for all } i = 5, \ldots, n+1$$

It is left to show that there exists a solution to this system of linear equations such that $0 < t_1 < \ldots < t_n < 1$.

First, note that all of the biases $b_1, \ldots, b_n, d$ can be written in terms of $t_1, \ldots, t_n$. Note that if we subtract $(R_4)$ from $(R_3)$, we can express $b_3$ in terms of the $t_i$ variables. The remaining equations become triangular, and therefore given any values for $t_i$'s we can back-substitute the remaining bias variables.

The same subtraction yields $w_3$ in terms of $t_i$'s. However, both $(R_1)$ and $(R_3) - (R_4)$ define $w_3$ in terms of the $t_i$ variables, so they must be the same:

$$\frac{1}{t_1} = \frac{1}{t_3 - t_2} + \frac{1}{t_4 - t_3}.$$

If we find values for $t_i$'s satisfying this equation and $0 < t_1 < \ldots < t_n < 1$, all other weights can be obtained by back-substitution since eliminating $w_3$ yields a triangular set of equations.

In particular, the following values are valid: $t_1 = \frac{1}{2n+1}$ and $t_i = \frac{2i-1}{2n+1}$ for all $i = 2, \ldots, n$. The remaining weights and biases can be obtained as described above, which completes the desired construction.

As an example, a construction with four units is depicted in Fig. 6. Its breakpoints are $t_1 = \frac{1}{9}$, $t_2 = \frac{3}{9}$, $t_3 = \frac{5}{9}$, and $t_4 = \frac{7}{9}$. Its ReLUs are $h_1(x) = \max\{0, -\frac{27}{2}x + \frac{3}{2}\}$, $h_2(x) = \max\{0, 9x - 3\}$, $h_3(x) = \max\{0, 9x - 5\}$, and $h_4(x) = \max\{0, 9x\}$. Finally, $\tilde{h}(x) = (-1, 1, -1, 1)^\top (h_1(x), h_2(x), h_3(x), h_4(x)) + 5$.

$\square$

## F   PROOF OF THEOREM 6

**Theorem 6.** *The maximal number of linear regions induced by a rectifier network with $n_0$ input units and $L$ hidden layers with $n_l \geq 3n_0$ for all $l$ is lower bounded by*

$$\left( \prod_{l=1}^{L-1} \left( \left\lfloor \frac{n_l}{n_0} \right\rfloor + 1 \right)^{n_0} \right) \sum_{j=0}^{n_0} \binom{n_L}{j}.$$

*Proof.* We follow the proof of Theorem 5 from (Montúfar et al., 2014) except that we use a different 1-dimensional construction. The main idea of the proof is to organize the network into $n_0$ independent networks with input dimension 1 each and apply the 1-dimensional construction to each individual network. In particular, for each layer $l$ we assign $\lfloor n_l/n_0 \rfloor$ ReLUs to each network, ignoring any remainder units. In (Montúfar et al., 2014), each of these networks have at least $\prod_{l=1}^{L} \lfloor n_l/n_0 \rfloor$ regions. We instead use Theorem 5 to attain $\prod_{l=1}^{L} (\lfloor n_l/n_0 \rfloor + 1)$ regions in each network.

Since the networks are independent from each other, the number of activation patterns of the compound network is the product of the number of activation patterns of each of the $n_0$ networks. Hence, the same holds for the number of regions. Therefore, the number of regions of this network is at least $(\prod_{l=1}^{L}(\lfloor n_l/n_0 \rfloor + 1))^{n_0}$.

In addition, we can replace the last layer by a function representing an arrangement of $n_L$ hyperplanes in general position that partitions $(0,1)^{n_0}$ into $\sum_{j=0}^{n_0} \binom{n_L}{j}$ regions. This yields the lower bound of $\prod_{l=1}^{L-1} (\lfloor n_l/n_0 \rfloor + 1)^{n_0} \sum_{j=0}^{n_0} \binom{n_L}{j}$.

$\square$

## G    PROOF OF THEOREM 7

**Theorem 7.** *For any values of $m \geq 1$ and $w \geq 2$, there exists a rectifier network with $n_0$ input units and $L$ hidden layers of size $2m + w(L-1)$ that has $2 \sum_{j=0}^{n_0-1} \binom{m-1}{j}(w+1)^{L-1}$ linear regions.*

*Proof.* Theorem 6.1 and Lemma 6.2 in Arora et al. (2016) imply that for any $m \geq 1$, we can construct a layer representing a function from $\mathbb{R}^n$ to $\mathbb{R}$ with $2m$ ReLUs that has $2 \sum_{j=0}^{n_0-1} \binom{m-1}{j}$ regions. Consider the network where this layer is the first one and the remaining layers are the one-dimensional layers from Theorem 5, each of size $w$. Then this network has size $2m + w(L-1)$ and $2 \sum_{j=0}^{n_0-1} \binom{m-1}{j}(w+1)^{L-1}$ regions. $\square$

## H    PROOF OF THEOREM 8

**Theorem 8.** *Consider a deep neural network with $L$ layers, $n_l$ rank-$k$ maxout units at each layer $l$, and an input of dimension $n_0$. The maximal number of regions of this neural network is at most*

$$\prod_{l=1}^{L} \sum_{j=0}^{d_l} \binom{\frac{k(k-1)}{2} n_l}{j}$$

*where $d_l = \min\{n_0, n_1, \ldots, n_l\}$.*

*Asymptotically, if $n_l = n$ for all $l = 1, \ldots, L$, $n \geq n_0$, and $n_0 = O(1)$, then the maximal number of regions is at most $O((k^2 n)^{L n_0})$.*

*Proof.* We denote by $W_j^l$ the $n_l \times n_{l-1}$ matrix where the rows are given by the $j$-th weight vectors of each rank-$k$ maxout unit at layer $l$, for $j = 1, \ldots, k$. Similarly, $b_j^l$ is the vector composed of the $j$-th biases at layer $l$.

In the case of maxout, an activation pattern $\mathcal{S} = (S^1, \ldots, S^l)$ is such that $S^l$ is a vector that maps from layer-$l$ neurons to $\{1, \ldots, k\}$. We say that the activation of a neuron is $j$ if $w_j x + b_j$ attains the maximum among all of its functions; that is, $w_j x + b_j \geq w_{j'} x + b_{j'}$ for all $j' = 1, \ldots, j$. In the case of ties, we assume the function with lowest index is considered as its activation.

Similarly to the ReLU case, denote by $\phi_{S^l} : \mathbb{R}^{n_l \times n_{l-1} \times k} \to \mathbb{R}^{n_l \times n_{l-1}}$ the operator that selects the rows of $W_1^l, \ldots, W_k^l$ that correspond to the activations in $S^l$. More precisely, $\phi_{S^l}(W_1^l, \ldots, W_k^l)$ is a matrix $W$ such that its $i$-th row is the $i$-th row of $W_j^l$, where $j$ is the neuron $i$'s activation in $S^l$. This essentially applies the maxout effect on the weight matrices given an activation pattern.

Montúfar et al. (2014) provides an upper bound of $\sum_{j=0}^{n_0} \binom{k^2 n}{j}$ for the number of regions for a single rank-$k$ maxout layer with $n$ neurons. The reasoning is as follows. For a single maxout unit,

there is one region per linear function. The boundaries between the regions are composed by pieces that are each contained in a hyperplane. Each piece is part of the boundary of at least two regions and conversely each pair of regions corresponds to at most one piece. Extending these pieces into hyperplanes cannot decrease the number of regions. Therefore, if we now consider $n$ maxout units in a single layer, we can have at most the number of regions of an arrangement of $k^2n$ hyperplanes. In the results below we replace $k^2$ by $\binom{k}{2}$, as only pairs of distinct functions need to be considered.

We need to define more precisely these $\binom{k}{2}n$ hyperplanes in order to apply a strategy similar to the one from the Section 3.1. In a single layer setting, they are given by $w_jx + b_j = w_{j'} + b_{j'}$ for each distinct pair $j, j'$ within a neuron. In order to extend this to multiple layers, consider a $\binom{k}{2}n_l \times n_{l-1}$ matrix $\hat{W}_l$ where its rows are given by $w_j - w_{j'}$ for every distinct pair $j, j'$ within a neuron $i$ and for every neuron $i = 1, \ldots, n_l$. Given a region $\mathcal{S}$, we can now write the weight matrix corresponding to the hyperplanes described above: $\hat{W}_\mathcal{S}^l := \hat{W}^l\, \phi_{S^{l-1}}(W_1^{l-1}, \ldots, W_k^{l-1}) \cdots \phi_{S^1}(W_1^1, \ldots, W_k^1)$. In other words, the hyperplanes that extend the boundary pieces within region $\mathcal{S}$ are given by the rows of $\hat{W}_\mathcal{S}^l x + b = 0$ for some bias $b$.

A main difference between the maxout case and the ReLU case is that the maxout operator $\phi$ does not guarantee reductions in rank, unlike the ReLU operator $\sigma$. We show the analogous of Lemma 3 for the maxout case. However, we fully relax the rank.

**Lemma 18.** *The number of regions induced by the $n_l$ neurons at layer $l$ within a certain region $\mathcal{S}$ is at most $\sum_{j=0}^{d_l} \binom{\frac{k(k-1)}{2}n_l}{j}$, where $d_l = \min\{n_0, n_1, \ldots, n_l\}$.*

*Proof.* For a fixed region $\mathcal{S}$, an upper bound is given by the number of regions of the hyperplane arrangement corresponding to $\hat{W}_\mathcal{S}^l x + b = 0$ for some bias $b$. The rank of $\hat{W}_\mathcal{S}^l$ is upper bounded by

$$
\begin{aligned}
\operatorname{rank}(\hat{W}_\mathcal{S}^l) &= \operatorname{rank}(\hat{W}^l\, \phi_{S^{l-1}}(W_1^{l-1}, \ldots, W_k^{l-1}) \cdots \phi_{S^1}(W_1^1, \ldots, W_k^1)) \\
&\leq \min\{\operatorname{rank}(\hat{W}^l), \operatorname{rank}(\phi_{S^{l-1}}(W_1^{l-1}, \ldots, W_k^{l-1})), \ldots, \operatorname{rank}(\phi_{S^1}(W_1^1, \ldots, W_k^1))\} \\
&\leq \min\{n_0, n_1, \ldots, n_l\}.
\end{aligned}
$$

Applying Lemma 2 yields the result. □

Since we can consider the partitioning of regions independently from each other, Lemma 18 implies that the maximal number of regions of a rank-$k$ maxout network is at most $\prod_{l=1}^{L} \sum_{j=0}^{d_l} \binom{\frac{k(k-1)}{2}n_l}{j}$ where $d_l = \min\{n_0, n_1, \ldots, n_l\}$.

□

## I  PROOF OF THEOREM 9

**Theorem 9.** *Provided that $|w_i^l h_j^{l-1} + b_i^l| \leq M$ for any possible value of $h^{l-1}$, a formulation with the set of constraints (1) for each neuron of a rectifier network is such that a feasible solution with a fixed value for $x$ yields the output $y$ of the neural network.*

*Proof.* For ease of explanation, we expand the set of constraints (1) as follows:

$$
W_i^l h_j^{l-1} + b_i^l = h_i^l - \overline{h}_i^l \tag{2}
$$

$$
h_i^l \leq M z_i^l \tag{3}
$$

$$
\overline{h}_i^l \leq M(1 - z_i^l) \tag{4}
$$

$$
h_i^l \geq 0 \tag{5}
$$

$$
\overline{h}_i^l \geq 0 \tag{6}
$$

$$
z_i^l \in \{0, 1\} \tag{7}
$$

It suffices to prove that the constraints for each neuron map the input to the output in the same way that the neural network would. If $W_i^l \mathbf{h}^{l-1} + b_i^l > 0$, it follows that $h_i^l - \overline{h}_i^l > 0$ according to (2). Since both variables are non-negative due to (5) and (6) whereas one is non-positive due to (3), (4), and (7), then $z_i^l = 1$ and $h_i^l = \max\left\{0, W_i^l \mathbf{h}^{l-1} + b_i^l\right\}$. If $W_i^l \mathbf{h}^{l-1} + b_i^l < 0$, then it similarly follows that $h_i^l - \overline{h}_i^l < 0$, $z_i^l = 0$, and thus $\overline{h}_i^l = \min\left\{0, W_i^l \mathbf{h}_j^{l-1} + b_i^l\right\}$. If $\sum_j W_i^l \mathbf{h}_j^{l-1} + b_i^l = 0$, then either $h_i^l = 0$ or $\overline{h}_i^l = 0$ due to constraints (5) to (7) whereas (2) implies that $\overline{h}_i^l = 0$ or $h_i^l = 0$, respectively. In this case, the value of $z_i^l$ is arbitrary but irrelevant. □

## J  EXACT COUNTING FOR RECTIFIER NETWORKS USING A MIXED-INTEGER FORMULATION

A systematic method to count these solutions is the one-tree approach (Danna et al., 2007), which resumes the search after an optimal solution has been found using the same branch-and-bound tree. That method can also be applied to near-optimal solutions by revisiting nodes pruned when solving for an optimal solution. Note that in constraints (1), the variables $z_i^l$ can be either 0 or 1 when they lie on the activation boundary, whereas we want to consider a neuron active only when its output is strictly positive. This discrepancy may cause double-counting when activation boundaries overlap. We can address that by defining an objective function that maximizes the minimum output $f$ of an active neuron, which is positive in non-degenerate cases. The formulation is as follows:

$$
\begin{aligned}
\max \quad & f \\
\text{s.t.} \quad & (1) && \text{for each neuron } i \text{ in layer } l && (8) \\
& f \le h_i^l + (1 - z_i^l)M && \text{for each neuron } i \text{ in layer } l \\
& x \in X
\end{aligned}
$$

**Corollary 19.** *The number of $z$ assignments of* (8) *yielding a positive objective function value corresponds to the number of linear regions of the neural network.*

*Proof.* Implicit in the discussion above. □

**Corollary 20.** *If the input $X$ is a polytope, then $(x, y)$ is mixed-integer representable.*

*Proof.* Immediate from the existence of a mixed-integer formulation mapping $x$ to $y$, which is correct as long as the input is bounded and therefore a sufficiently large $M$ exists. □

In practice, the value of constant $M$ should be chosen to be as small as possible, which also implies choosing different values on different places to make the formulation tighter and more stable numerically (Camm et al., 1990). For the constraints set (1), it suffices to choose $M$ to be as large as either $h_i^l$ or $\overline{h}_i^l$ can be given the bounds on the input. Hence, we can respectively replace $M$ with $H_i^l$ and $\bar{H}_i^l$ in the constraints involving those variables. If we are given lower and upper bounds for $X$, which we can use for $H^0$ and $\bar{H}^0$, then we can define subsequent bounds as follows:

$$
H_i^l = \max\left\{0, \sum_j \max\left\{0, w_{ij}^l H_j^{l-1}\right\} + b_i^l\right\}
$$

$$
\overline{H}_i^l = \max\left\{0, \sum_j \max\left\{0, -w_{ij}^l H_j^{l-1}\right\} - b_i^l\right\}
$$

For the constraint involving $f$ in formulation (8), we should choose a slightly larger value than $H_i^l$ for correctness because some neurons may never be active within the input bounds.

# K   COUNTING LINEAR REGIONS OF RELUS WITH UNRESTRICTED INPUTS

More generally, we can represent linear regions as a disjunctive program (Balas, 1979), which consist of a union of polyhedra. Disjunctive programs are used in the integer programming literature to generate cutting planes by lift-and-project (Balas et al., 1993). In what follows, we assume that a neuron can be either active or inactive when the output lies on the activation hyperplane.

For each active neuron, we can use the following constraints to map input to output:

$$w_i^l h^{l-1} + b_i^l = h_i^l \tag{9}$$

$$h_i^l \geq 0 \tag{10}$$

For each inactive neuron, we use the following constraint:

$$w_i^l h^{l-1} + b_i^l \leq 0 \tag{11}$$

$$h_i^l = 0 \tag{12}$$

**Theorem 21.** *The set of linear regions of a rectifier network is a union of polyhedra.*

*Proof.* First, the activation set $S^l$ for each level $l$ defines the following mapping:

$$\bigcup_{S^l \subseteq \{1,\ldots,n_l\}, l \in \{1,\ldots,L+1\}} \left\{ (h^0, h^1, \ldots, h^{L+1}) \mid (9)-(10) \text{ if } i \in S^l; (11)-(12) \text{ otherwise} \right\} \tag{13}$$

Consequently, we can project the variables sets $h^1, \ldots, h^{L+1}$ out of each of those terms by Fourier-Motzkin elimination (Fourier, 1826), thereby yielding a polyhedron for each combination of active sets across the layers. □

Note that the result above is similar in essence to Theorem 2 of Raghu et al. (2017).

**Corollary 22.** *If $X$ is unrestricted, then the number of linear regions can be counted using (8) if $M$ is large enough.*

*Proof.* To count regions, we only need one point $x$ from each linear region. Since the number of linear regions is finite, then it suffices if $M$ is large enough to correctly map a single point in each region. Conversely, each infeasible linear region either corresponds to empty sets of (13) or else to a polyhedron $P$ such that $\{(h^1, \ldots, h^{L+1}) \in P \mid h_i^l > 0 \ \forall l \in \{1, \ldots, L+1\}, i \in S^l\}$ is empty, and neither case would yield a solution for the $z$-projection of (8). □

# L   MIXED-INTEGER REPRESENTABILITY OF MAXOUT UNITS

In what follows, we assume that we are given a neuron $i$ in level $l$ with output $h_i^l$. For that neuron, we denote the vector of weights as $w_1^{li}, \ldots, w_k^{li}$. Thus, the neuron output corresponds to

$$h_i^l := \max \left\{ w_1^{li} h^{l-1} + b_1, \ldots, w_k^{li} h^{l-1} + b_k \right\}$$

Hence, we can connect inputs to outputs for that given neuron as follows:

$$w_j^{li} h_j^{l-1} + b_j^{li} = g_j^{li}, \qquad j = 1, \ldots, k \tag{14}$$

$$h_i^l \geq g_j^{li}, \qquad j = 1, \ldots, k \tag{15}$$

$$h_i^l \leq g_j^{li} + M(1 - z_j^{li}) \qquad j = 1, \ldots, k \tag{16}$$

$$z_j^{li} \in \{0, 1\}, \qquad j = 1, \ldots, k \tag{17}$$

$$\sum_{j=1}^{k} z_j^{li} = 1 \tag{18}$$

The formulation above generalizes that for ReLUs with some small modifications. First, we are computing the output of each term with constraint (14). The output of the neuron is lower bounded by that of each term with constraint (15). Finally, we have a binary variable $z_m^{li}$ per term of each neuron, which denotes which neuron is active. Constraint (18) enforces that only one variable is at one per neuron, whereas constraint (16) equates the output of the neuron with the active term. Each constant $M$ should be chosen in a way that the other terms can vary freely, hence effectively disabling the constraint when the corresponding binary variable is at zero.

## M  ACCURACY AND ERROR MEASURES OF THE SAMPLE NETWORKS

Figure 8 shows the error during training for different configurations in the first experiment. Figure 9 shows the errors after training for different configurations in the second experiment. In both, we observe some relation between accuracy and the order of magnitude of the linear regions, which suggest that linear regions represent a reasonable proxy to the representational power of DNNs.

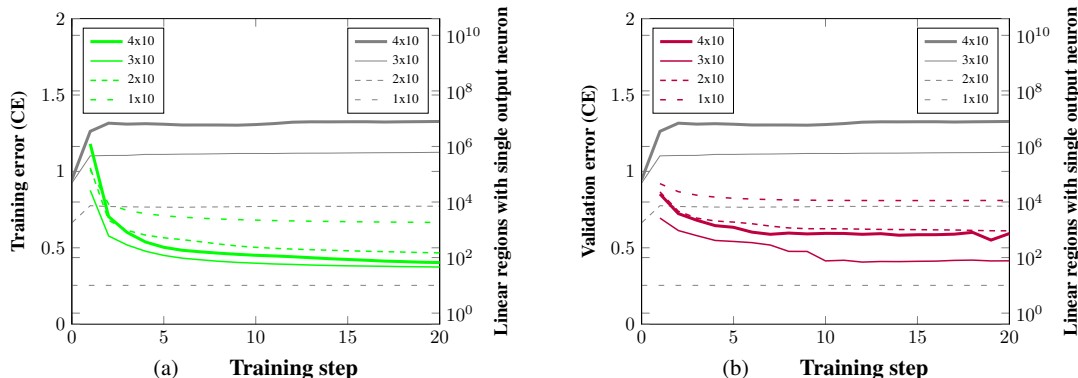

Figure 8: *Contrast of cross-entropy along training with number of regions identifying a single digit in the first experiment: (a) shows training error in green; (b) shows validation error in purple.*

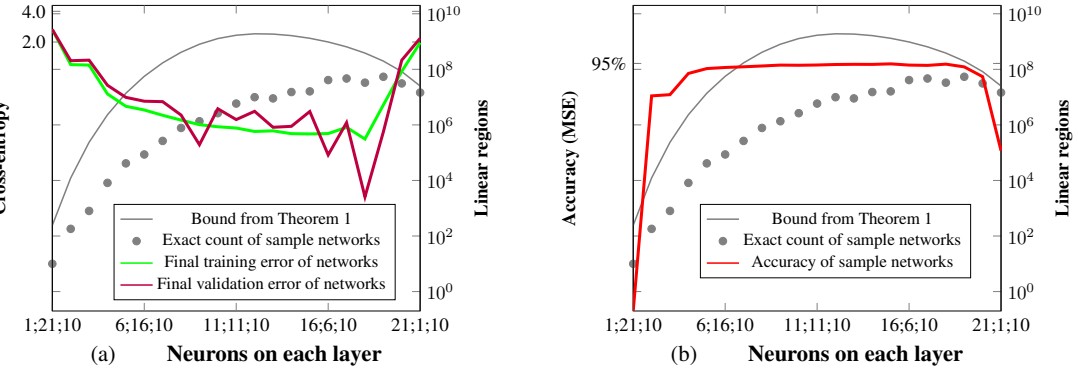

Figure 9: *Contrast of final errors with number of regions and bound in the second experiment: (a) shows training error in green and validation error in purple; (b) shows accuracy in red.*

## N    RUNTIMES FOR COUNTING THE LINEAR REGIONS

Table 1 reports the runtimes to count different configurations of networks on each experiment.

| Experiment | Network widths | Runtime (s) |
| --- | --- | --- |
| 1 | $1 \times 10$ | $6.0 \times 10^{-2}$ |
| | $2 \times 10$ | $1.1 \times 10^2$ |
| | $3 \times 10$ | $1.8 \times 10^3$ |
| | $4 \times 10$ | $5.2 \times 10^4$ |
| 2 | $1; 21; 10$ | $1.0 \times 10^{-2}$ |
| | $2; 20; 10$ | $4.5 \times 10^{-1}$ |
| | $3; 19; 10$ | $1.9 \times 10^0$ |
| | $4; 18; 10$ | $3.8 \times 10^1$ |
| | $5; 17; 10$ | $2.0 \times 10^2$ |
| | $6; 16; 10$ | $4.1 \times 10^2$ |
| | $7; 15; 10$ | $1.2 \times 10^3$ |
| | $9; 13; 10$ | $7.5 \times 10^3$ |
| | $10; 12; 10$ | $1.5 \times 10^4$ |
| | $11; 11; 10$ | $3.3 \times 10^4$ |
| | $12; 10; 10$ | $4.4 \times 10^4$ |
| | $13; 9; 10$ | $5.8 \times 10^4$ |
| | $14; 8; 10$ | $6.6 \times 10^4$ |
| | $15; 7; 10$ | $7.5 \times 10^4$ |
| | $16; 6; 10$ | $3.0 \times 10^5$ |
| | $17; 5; 10$ | $2.8 \times 10^5$ |
| | $18; 4; 10$ | $2.3 \times 10^5$ |
| | $19; 3; 10$ | $2.7 \times 10^5$ |
| | $20; 2; 10$ | $1.1 \times 10^5$ |
| | $21; 1; 10$ | $4.0 \times 10^4$ |

Table 1: Runtimes for counting the trained networks for each configuration used in the experiments.

## O    UPPER BOUND BY VARYING THE TOTAL NUMBER OF NEURONS

Figure 10a shows that the upper bound from Theorem 1 can only be maximized if more layers are added as the number of neurons increase. In contrast, Figure 10b shows that the smallest depth preserving such growth is better because there is a secondary, although still exponential, effect that starts shrinks the bound if the number of layers is too large for the total number of neurons.

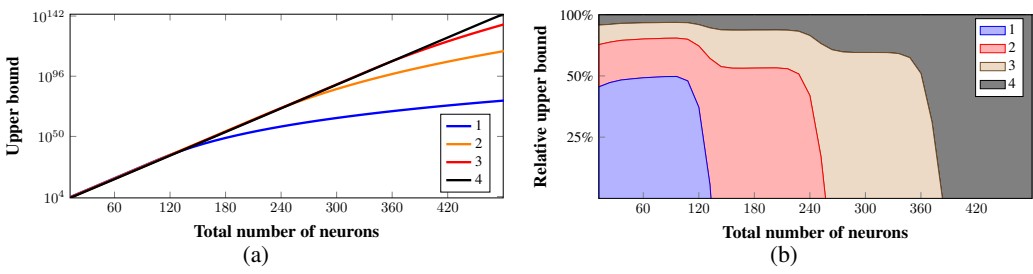

Figure 10: Bounds from Theorem 1 in semilog scale for $n_0 = 60$ as the total number of neurons increase by evenly distributing such neurons in 1 to 4 layers: (a) actual values showing overall impact of more depth; and (b) ratio by sum over all layers showing local impact of particular depths.

