# OpenReview forum: "Bounding and Counting Linear Regions of Deep Neural Networks"
_ICLR.cc/2018/Conference — Reject_

### Official Review · AnonReviewer1 · 2017-11-27
**The paper builds on previous work to bound and count the number of linear regions in ReLU networks, and evaluates this in small experiments.**

**Rating:** 6
**Confidence:** 5

**Review:**

This paper investigates the complexity of neural networks with piecewise linear activations by studying the number of linear regions of the representable functions. It builds on previous works Montufar et al. (2014) and Raghu et al. (2017) and presents improved bounds on the maximum number of linear regions. It also evaluates the number of regions of small networks during training.

The improved upper bound given in Theorem 1 appeared in SampTA 2017 - Mathematics of deep learning ``Notes on the number of linear regions of deep neural networks'' by Montufar.

The improved lower bound given in Theorem 6 is very modest but neat. Theorem 5 follows easily from this.

The improved upper bound for maxout networks follows a similar intuition but appears to be novel.

The paper also discusses the exact computation of the number of linear regions in small trained networks. It presents experiments during training and with varying network sizes. These give an interesting picture, consistent with the theoretical bounds, and showing the behaviour during training.

Here it would be interesting to run more experiments to see how the number of regions might relate to the quality of the trained hypotheses.

---

> ### Author Response · Authors · 2017-12-09
> **Preliminary feedback**
>
> We are currently working on addressing all the comments of the reviewers. However, we would like to provide a brief update on the status and current progress on our part in addressing the concerns.
>
> 1) "The improved upper bound given in Theorem 1 appeared in SampTA 2017 - Mathematics of deep learning "Notes on the number of linear regions of deep neural networks" by Montufar."
>
> We were not aware of the paper in our original submission and we searched for it but it is not available online. We have emailed the author for a copy. As soon as we obtain one, we will clarify the relationship between both papers.

---

### Official Review · AnonReviewer2 · 2017-11-27

**Rating:** 4
**Confidence:** 5

**Review:**

Paper Summary:

This paper looks at providing better bounds for the number of linear regions in the function represented by a deep neural network. It first recaps some of the setting: if a neural network has a piecewise linear activation function (e.g. relu, maxout), the final function computed by the network (before softmax) is also piecewise linear and divides up the input into polyhedral regions which are all different linear functions. These regions also have a correspondence with Activation Patterns, the active/inactive pattern of neurons over the entire network. Previous work [1], [2], has derived lower and upper bounds for the number of linear regions that a particular neural network architecture can have. This paper improves on the upper bound given by [2] and the lower bound given by [1]. They also provide a tight bound for the one dimensional input case. Finally, for small networks, they formulate finding linear regions as solving a linear program, and use this method to compute the number of linear regions on small networks during training on MNIST

Main Comments:
The paper is very well written and clearly states and explains the contributions. However, the new bounds proposed (Theorem 1, Theorem 6), seem like small improvements over the previously proposed bounds, with no other novel interpretations or insights into deep architectures. (The improvement on Zaslavsky's theorem is interesting.) The idea of counting the number of regions exactly by solving a linear program is interesting, but is not going to scale well, and as a result the experiments are on extremely small networks (width 8), which only achieve 90% accuracy on MNIST. It is therefore hard to be entirely convinced by the empirical conclusions that more linear regions is better. I would like to see the technique of counting linear regions used even approximately for larger networks, where even though the results are an approximation, the takeaways might be more insightful.

Overall, while the paper is well written and makes some interesting points, it presently isn't a significant enough contribution to warrant acceptance.

[1] On the number of linear regions of Deep Neural Networks, 2014, Montufar, Pascanu, Cho, Bengio
[2] On the expressive power of deep neural networks, 2017, Raghu, Poole, Kleinberg, Ganguli, Sohl-Dickstein

---

> ### Author Response · Authors · 2017-12-09
> **Preliminary feedback**
>
> We are currently working on addressing all the comments of the reviewers. However, we would like to provide a brief update on the status and current progress on our part in addressing the concerns.
>
> 1) "[...] The new bounds proposed (Theorem 1, Theorem 6), seem like small improvements over the previously proposed bounds, with no other novel interpretations or insights into deep architectures."
>
> A novel interpretation derived from Theorem 1 is on the relationship between the number of linear regions and the widths of the layers of the DNN. We emphasize that in Theorem 1, the summations depend on the minimum width across the previous layers. This yields the insight that the number of regions is affected by a bottleneck-like effect from earlier layers. In other words, the bound from Theorem 1 is smaller if the earlier layers are smaller rather than if the later layers are smaller, fixed the total size of the network. This is reflected in the upper bound plot in Figure 4(b) and further validated by the computational results shown in the same figure.
>
> In addition, the insights behind Theorem 1 pave the road to Theorem 5, which exploits the dimensionality of the regions in order to achieve the exact maximal number of regions for the one-dimensional case.
>
> The case of more dimensions has proven to be more challenging, as evidenced by previous papers on the topic, but Theorem 6 nevertheless achieves a modest improvement. It generalizes the insight of Theorem 5 to higher dimensions.
>
> We will elaborate on this discussion in the paper.
>
>
> 2) "I would like to see the technique of counting linear regions used even approximately for larger networks, where even though the results are an approximation, the takeaways might be more insightful."
>
>
> We agree with the reviewer that more insight could be obtained with larger networks. However, exact counting has never been done before and we are excited about this new capability. While this is not fully scalable in the current form, this serves as a proof-of-concept that already provides insights even at a small scale.
>
> Nevertheless, as the reviewer correctly pointed out, moving towards larger networks may require approximations. While we have already been thinking about using approximations, this is a different line of research and may need substantial additional work.
>
>
> 3) "[...] as a result the experiments are on extremely small networks (width 8), which only achieve 90% accuracy on MNIST."
>
> In order to partially address this concern, we are working on counting (possibly larger) networks with higher accuracy.

---

### Official Review · AnonReviewer3 · 2017-11-29

**Rating:** 6
**Confidence:** 3

**Review:**

This is quite an interesting paper. Thank you. Here are a few comments:

I think this style of writing theoretical papers is pretty good, where the main text aims of preserving a coherent story while the technicalities of the proofs are sent to the appendix.
However I would have appreciated a little bit more details about the proofs in the main text (maybe more details about the construct that is involved). I can appreciate though that this a fine line to walk. Also in the appendix, please restate the lemma that is being proven. Otherwise one will have to scroll up and down all the time to understand the proof.

I think the paper could also discuss a bit more in detail the results provided. For example a discussion of how practical is the algorithm proposed for exact counting of linear regions would be nice. Though regardless, I think the findings speak for themselves and this seems an important step forward in understanding neural nets.

****************
I had reduced my score based on the observation made by Reviewer 1 regarding the talk Montufar at SampTA. Could the authors prioritize clarification to that point !
 - Thanks for the clarification and adding this citation.

---

> ### Author Response · Authors · 2017-12-09
> **Preliminary feedback**
>
> We are currently working on addressing all the comments of the reviewers. However, we would like to provide a brief update on the status and current progress on our part in addressing the concerns.
>
> 1) "[...] I would have appreciated a little bit more details about the proofs in the main text. [...] I think the paper could also discuss a bit more in detail the results provided."
>
> We will improve the discussion and move some of the contents from the Appendix to the main section.
>
>
> 2) "[...] observation made by Reviewer 1 regarding the talk Montufar at SampTA."
>
> Please see our answer to AnonReviewer1.

---

### Author Response · Authors · 2017-12-30
**Feedback after revising the manuscript**

We would like to specially thank the reviewers for constructive feedback and suggestions. This was extremely useful in revising the manuscript. We have tried to address all the concerns of the reviewers. In particular, we highlight the major changes in the revised submission.

1) We managed to get a copy of Montufar’s 2017 SampTA paper through email communication. We thank Reviewer 1 for pointing this out, and we did observe that Theorem 1 in our original submission is also shown in his paper. Montufar generally suggests that the bound can be improved by looking more closely at dimensions, but he does not provide a way to do it. We had addressed this in the submitted version by studying the rank of weight matrices. While it is unfortunate that we did not find an online copy of this paper during our submission, on the positive side, this pushed us to further improve our upper bound in the revised manuscript. In particular, our bound is tighter now and also produces additional novel insights when the input dimension is large. We hope that this will satisfy the main concerns of Reviewer 1 and Reviewer 3.

2) As suggested by Reviewer 2, we have added more intuitions for the theorems and lemmas in the paper. Using the revised bounds, we have made an interesting observation that is very different from previous results. In particular, the results in Montufar et al. 2014 show that, if the input dimension is constant, then the number of regions of deep networks is asymptotically larger than those of shallow (single-layer) networks. While most prior results assert the claim that more depth leads to better representational power, we have observed scenarios where shallow networks have larger number of linear regions, i.e., better representational power. Using our revised upper bound, we show that if the input dimension is large, then shallow networks have more regions than deep networks. More precisely, when the input dimension is higher than the total number of neurons, then a deep network of $L$ layers each having width $n$ has fewer linear regions compared to a shallow network that has a single layer of width $Ln$. Exact details are given in the paper and the appendix. Note that this result is particularly interesting, since it is different from prior results and cannot be obtained from earlier bounds.

3) To address the concerns of Reviewer 2, we performed several additional experiments on improved MNIST networks with accuracy closer to 95%. Please note that before this work, the idea of linear regions for deep neural networks is only a theoretical concept. Although we only count on simple networks such as MNIST, it nevertheless reinforces the idea that such theoretical ideas can be validated in real experiments. Unfortunately, the exact counting for larger networks is infeasible using the current approach, but improvements or workarounds could be devised. For instance, we would like to thank Reviewer 2 for suggesting the problem of “approximate counting” of linear regions. This is a promising approach and it can be done as future work, as finding good approximations involves a separate line of research. We would like to note that the counting procedure serves to validate the bound and that it could also provide further insight for future work on tighter bounds. If and when these bounds get close enough, it might not be relevant to do exact counting anymore.

4) As suggested by Reviewer 3, we have restated the lemmas/theorems in the Appendix. We have also added runtimes for the  exact counting of different networks in the Appendix.

5) As per the suggestion given by Reviewer 1, we now report and discuss experimental results on the relation between the number of linear regions and the quality of training. While we have anecdotal evidence from the plots that larger number of linear regions generally correspond to better accuracy, we believe that this requires a more thorough investigation. In particular, we believe that the quality of the training also depends on the shape of the linear regions, which is not represented by just the number of linear regions. Independently of the precise relationship, our procedure opens a new door to an extensive empirical investigation.

The paper length has increased to 9.5 pages (note that there is no actual page limit for this conference), but we could easily rearrange the contents based on the final recommendation of the reviewers and Area Chairs, if the paper gets accepted. We really appreciate your feedback and we would be happy to address any additional concerns that you may have.

---

### Author Response · Authors · 2018-01-05
**Last update**

We just uploaded a new version of the manuscript. We have backed some intuitions on the exponential growth of linear regions for large input dimension, improved the readability of the paper, and refined the discussion.

---

### Author Response · Authors · 2018-01-05
**Full list of changes since initial submission**

This is a full list of changes made to the paper, except for changes made for readability. The changes and their motivations are discussed in previous comments; this list is only for the convenience of the reviewers.

Below, Montufar2017 refers to Montufar's 2017 SampTA paper that contained the previous upper bound, and Arora2016 refers to Arora et al. 2016 (see reference in paper), which provides a different lower bound to the maximal number of regions.

1. Introduction

- Added to the discussion the Montufar2017 and Arora2016 papers.

2. Notations and background

- Before "Main contributions", all changes in this section are for readability, except for the addition of a citation to Montufar2017.

- We revise "Main contributions" in light of the overall changes to the paper, which are discussed in a previous comment. The main addition is the highlighting of the insights for the large input dimension case.

3. Tighter bounds for rectifier networks

- In the beginning, we add previous bounds from Montufar2017 and Arora2016.

3.1. An upper bound on the number of linear regions

- A major change is the bound in Theorem 1. We tighten the upper bound result by considering dimensions more precisely.

- We thoroughly discuss two insights from Theorem 1. One of them is the bottleneck effect, which was briefly considered in the first version. In this version, we discuss it more extensively, add some plots illustrating the effect, and prove supporting results in Appendix A. The second insight is the case where the input dimension is large, in which we compare shallow and deep networks. Once again, we discuss it extensively, provide plots, and prove results in Appendix A.

- Following these discussions, most of the ingredients of the proof of Theorem 1 are the same. In particular, we merge the first version's Lemmas 3 and 4 into the current version's Lemma 3, but with a slightly different result that leads to the new proof. The current Lemma 4 is part of the new proof, which is completed in Appendix D. We have moved some of the shorter proofs from the appendix to the main text for ease of readability.

3.2. The case of dimension one

- Readability changes only.

3.3. A lower bound on the maximal number of regions

- We extend a lower bound from Arora2016 with the one-dimensional construction, as it was similarly done for the Montufar et al. lower bound.

4. An upper bound on the number of linear regions for maxout networks

- Modified wording due to changes in its relationship with Theorem 1.

5. Exact counting of linear regions

- We include a reference to Cheng et al. 2017, which considers a MIP formulation for a DNN in a different context.

6. Experiments

- Both experiments are replaced by experiments on larger networks, with width 10 instead of 8 in the first experiment and total number of neurons 22 instead of 16 in the second experiment. The upper bound in the plot is updated with the new one.

- These networks approach test error of 6% in the first case and 5% in the second.

- We also provide training and final errors in Appendix M counting runtimes in Appendix N.

7. Discussion

- The discussion was extensively revised, including the following changes:

- We mention the finding on shallow networks with large input dimension.

- We point out that the new version of Theorem 1 has particular depths maximizing the upper bound according to the input dimension and the total number of neurons, which could be investigated for the actual number of regions. This is illustrated with an example in Appendix O.

- We also discuss other directions for future work in the last paragraph, including understanding the relation between number of regions, accuracy, and potential for overfitting.


Appendices

In the new version, we have added appendices A, B, G, M, N, and O. Appendix A elaborates on the properties of the new version of Theorem 1. Appendix B shows that the maximal number of linear regions is also exponential for large input dimensions. Appendix G describes how the lower bound from Arora2016 can be extended with the new one-dimensional construction. Appendices M and N provide error measures and runtimes for the experiments. Appendix O provides additional plots based on Theorem 1.

Other appendices have been changed: Appendix D (previously also D, Proof of Theorem 1) has been rewritten with the new proof. Appendix H (previously G, proof of the maxout upper bound) was slightly revised in light of the change in Theorem 1. Previous appendices B and C (in the first version) have been moved to the main text (and altered according to the new proof).

---

### Decision · Program_Chairs · 2018-01-29
**ICLR 2018 Conference Acceptance Decision**

**Decision:**

Reject

**Comment:**

Dear authors,

The reviewers appreciated your work and recognized the importance of theoretical work to understand the behaviour of deep nets. That said, the improvement over existing work (especially Montufar, 2017) is minor. This, combined with the limited attraction of such work, means that the paper will not be accepted.

I acknowledge the major modifications done but it is up to the reviewers to decide whether or not they agree to re-review a significantly updated version.